# Scenario Simulation of Land Use and Cover under Safeguarding Ecological Security: A Case Study of Chang-Zhu-Tan Metropolitan Area, China

Zhiwei Deng [1], Bin Quan [1,2,*], Haibo Zhang [1], Hongqun Xie [1] and Ze Zhou [1]

1 College of Geography and Tourism, Hengyang Normal University, Hengyang 421002, China; dzw17673290352@aliyun.com (Z.D.); zhb3909@hynu.edu.cn (H.Z.); hongqun1225@163.com (H.X.); zhouze1019@163.com (Z.Z.)
2 Hengyang Base of International Centre on Space Technologies for Natural and Cultural Heritage under the Auspices of UNESCO, Hengyang 421002, China
* Correspondence: quanbin308@aliyun.com

**Abstract:** Scenario-based simulation in land use and cover change (LUCC) is a practical approach to maintaining ecological security. Many studies generally set constraints of LUCC utilizing ecological patches but without consideration of corridors connecting these patches. Here, we constructed a framework to balance urban growth and ecological security by integrating ecological security patterns (ESPs) into the PLUS model. This study selected Chang-Zhu-Tan Metropolitan Area (CZTMA) in central China as a typical case. Specifically, coupling quantitative demand with spatial constraints of multiple levels of ESPs, this study designed four scenarios, including historical tendency (HT), urban growth (UG), ecological conservation (EC), and coordinating city development and ecological protection (CCE). Then, the transformations and landscape patterns of LUCC were analyzed to evaluate the future land change from 2020 to 2050. The results show sixty-one key ecological sources in the CZTMA, mainly in higher-elevation forested areas. Forty-six ecological corridors were estimated using circuit theory. The building expansion was driven by accessibility to transportation and government locations and will contribute to the loss of forest and cropland in the future. The feature of different scenarios in alleviating the increasing fragmentation of patches and reducing the loss amount of ecological land showed EC > CCE > HT > UG. This study developed the ESP-PLUS framework and its modeling idea, which has the potential to be applied in other regions. This extension would assist decision-makers and urban planners in formulating sustainable land strategies that effectively reconcile eco-environmental conservation with robust economic growth, achieving a mutually beneficial outcome.

**Keywords:** land use and cover change; ecological security pattern; multiple scenarios simulation; PLUS model; Chang-Zhu-Tan metropolitan area



## 1. Introduction

The global land use and cover change (LUCC) has become significant due to the progress of industrialization, urbanization, and the increased intensity of human activities [1,2]. It leads to issues that threaten human well-being, such as wetland loss, land degradation, climate extremes, and biodiversity reduction [3–6]. Previous research showed that about 60% of global land change directly relates to human activities [7]. Meanwhile, the global impervious surface increased by 480 thousand $km^2$ while the forest decreased by 1.47 million $km^2$ during the past four decades [8]. The LUCC is more prominent in metropolitan areas than other regions driven by population agglomeration from rural to urban areas and rapid economic growth, which puts enormous pressure on the local ecological environment [9,10]. Therefore, simulating and understanding LUCC in metropolitan

regions and their potential trends from ecological security is essential for coordinating socioeconomic development and ecological conservation.

Predicting the spatiotemporal processes of LUCC is recognized as a helpful approach to supporting urban planning and optimizing land use allocations [11]. The primary differences among models that simulate the LUCC dynamics are spatial and non-spatial. The non-spatial models primarily emphasize the simulation of change in amount and speed with relatively little focus on space operation. Examples of such models include multi-objective programming (MOP) [12], grey-system models (GM) [13], system dynamics (SD) [14], and Markov chains (MC) [15]. The utilization of cellular automata (CA) has gained significant recognition in geographical modeling and parallel data processing, primarily attributed to its robust capabilities [16]. Nevertheless, the single model does possess several limitations despite its inherent benefits. As a typical coupling model, CA-Markov is broadly employed to explore future land use/cover evolution [17–19] because it can effectively absorb the benefits of time series forecast and spatial projection of CA and MC methods.

To better determine the transition rules in land change modeling, scientists have proposed admirable models and algorithms based on CA, such as multi-criteria evaluation (MCE) [20], logistic regression (LR) [21], multiple layer perception (MLP)-ANN [22], GEOMOD [23], the SLEUTH model [24], conversion of land use and its effects (CLUE) family models [25–27], and the future land use simulation (FLUS) model [28]. However, most available models are weak in detecting the driving factor's contribution to land use change and capturing the evolutionary rules of the multiple-type patches. Liang et al. [29] recently introduced a patch-generating land use simulation (PLUS) model. This model incorporates enhancements to the adaptable inertia and roulette competition processes shown in the FLUS model and integrates the random forest method. Relevant evidence suggests that the PLUS model can deal with multicollinearity in driving factors and simulate the patterns of LUCC more accurately than the previous models could [30–32]. The model has superior performance among simulation results in Rwanda [33], the ecological area of western Beijing [34], the Shiyang watersheds of arid regions [35], and the Qinghai–Tibet Plateau [36]. Thus, adopting the PLUS model and verifying its suitability in metropolitan areas could provide new thoughts for the LUCC simulation in other parts of the world. Moreover, it is essential to model the LUCC considering ecological security when urban growth is inevitable [37].

In simulating land changes, natural reserves [38], biological habitats [39], ecological red-line regions [40], and areas where ecosystem services work well [41] were used as constraints to optimize land use patterns and control urban sprawl. However, those approaches remain issues that deserve further exploration. Firstly, the ecological patches of those constraints are isolated and do not constitute a structured ecological network system. Corridors as belt regions in ecological security patterns are crucial for connecting material, energy, and information flows, especially for animal migrations [42]. Consequently, incorporating ecological corridors into the constraints can fill this deficiency. Secondly, most of the previous ecological constraints are relatively single, which cannot match land demand under multiple scenarios of LUCC [43]. The quantitative land use/cover demands for various scenarios could be designed by adjusting the future transition probabilities across land categories based on the MC approach. Hence, the spatial constraints could be diverse rather than unchanging when modelers develop multiple scenarios.

With urbanization and industrialization, the demands of human activities on land resources are increasing, especially in developing metropolitan areas [44]. Although metropolitans have achieved remarkable progress in socioeconomic areas, unprecedented transformations in land use and cover have occurred and resulted in resources and environmental problems, such as declining air quality [45], extreme traffic by congestion [46], high potential in soil erosion risk [47], lowered water tables [48], urban waterlogging [49], deforestation [50], and occupied cropland [51]. At the same time, these consequences restricted the transition of socioeconomic areas and reduced the level of regional ecological

security [52]. Therefore, studying LUCC in metropolitan areas is meaningful for achieving ecological civilization and healthy urban development. The Chang-Zhu-Tan urban agglomeration became a comprehensive reform zone for China's resource-saving and environmentally friendly society in 2007 [53]. Since then, it has received more support in national policy and entered a stage of rapid economic advancement. However, there are still significant gaps in the overall level of economic development compared to other developed cities. It is challenging to drive the development of neighboring cities [54]. Hence, in February 2022, the Chinese Government set up the Chang-Zhu-Tan Metropolitan Area (CZTMA) to promote sustainable development throughout the region by accelerating the integration of the three cities (Changsha, Zhuzhou, and Xiangtan) in space, industry, and resource [55].

The CZTMA, as the growth core of Hunan Province's economic development, plays a crucial role in promoting the rise of central China. However, the land use and cover in the CZTMA have remarkably changed due to human disturbance and the urgent need for economic growth [56]. Furthermore, climate change and extreme weather events such as typhoons, floods, and droughts threaten the regional ecological security [57,58]. In the previous studies on the Chang-Zhu-Tan region, the analytical perspectives of land change focused on quantifying land transformation [59], land use carbon emissions [60], eco-environmental quality assessment [55], and urban boundary delineation [61]. Moreover, scientists examined ecological network evolution at small scales [62]. They predicted temporal future land changes under coordinated scenarios from urban development and ecological protection [53], involving parts of the CZTMA, which overlooked the simultaneous quantitative and spatial conservation for ecological resources. Meanwhile, the studies on multi-scenario simulation of LUCC incorporating ESPs in the CZTMA are still lacking. Thus, we formulated the two research questions of this study as follows:

- How to frame a predictive modeling for LUCC that considers ecological security?
- What are the LUCC dynamics and their enlightenment under the framework?

Accordingly, we selected the CZTMA as a typical case. This research aimed to model a technical framework to forecast the LUCC trend via combining the PLUS model with the multi-threshold ESPs, evaluate and compare the future LUCC and landscape characteristics, and explore diverse spatial variables' impacts on the LUCC. This study can provide informative references and modeling ideas for guiding orderly cities' growth and sustaining the overall safety of regional ecosystems in metropolitan areas.

## 2. Materials and Methods

### 2.1. Study Area

Since 2021, the Chinese government has established seven national-level metropolitan areas to promote industrial and population concentration and high-quality socioeconomic development. The Chang-Zhu-Tan Metropolitan Area (CZTMA) was designed to integrate the three cities to generate stronger economic radiation and accelerate the growth of underdeveloped regions [55]. The CZTMA is situated in Hunan Province's north-central region (111°53′–114°16′ E, 27°13′–28°40′ N). The cities serve as the economic, cultural, and industrial cores of Hunan Province, playing a significant role in driving the achievement of high-quality development in central China. The extent of this study encompasses the complete geographical region of Changsha City and selected portions of Zhuzhou City and Xiangtan City, with an area of roughly 18,900 km$^2$ (Figure 1).

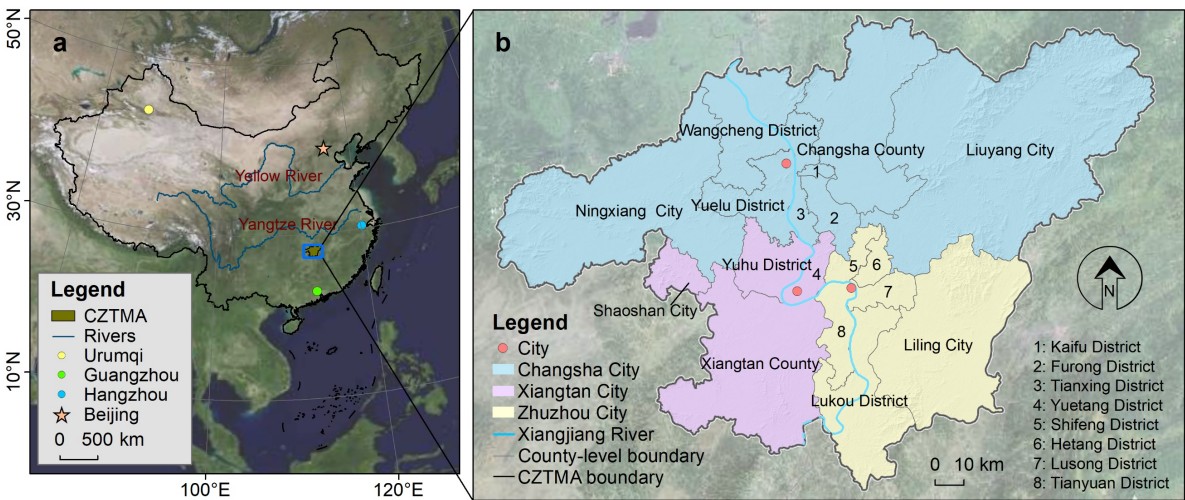

**Figure 1.** Location (**a**) and administrative divisions (**b**) of the CZTMA in China.

The study area's population grew from 12.39 million in 2000 to 16.99 million in 2022, and its GDP reached 2.57 hundred billion dollars [63]. The intercity trains run across the region, and the Beijing–Guangzhou, Shanghai–Kunming, and Chongqing–Xiamen high-speed railways come together here. The climatic conditions observed in the research area can be classified as subtropical monsoon, which is characterized by an average annual temperature ranging from 16 to 18 °C and an annual precipitation ranging from 1200 to 1500 mm. Here, imbalanced rainfall can cause floods or droughts. The Xiangjiang River, flowing from south to north, is a first-class part of the Yangtze River. The region's topography is characterized by the Luoxiao Mountains in the east, hills in the west, and plains along the river in the center.

### 2.2. Data Source and Processing

Several models and tools are used in this study, and they require different data formats and types. The Resources and Environmental Sciences Data Center of the Chinese Academy of Sciences (http://www.resdc.cn/, accessed on 11 May 2022) provides land use/cover data from 2000, 2010, and 2020. These data rely on multi-temporal remote sensing imagery from Landsat TM/ETM+/OLI and are then interpreted through human–machine interaction with a spatial resolution of 30 m. It has 6 first-classes and 25 subclasses, among which the accuracy of first-class interpretation is above 85% [64,65]. We employed the data product's first-level classification, which comprises cropland, forest, grassland, waters, buildings, and unused land (Table 1). The unused land with a small area was classified as grassland, according to the observation of Google Earth images. Finally, we divided land categories into cropland, forest, grassland, waters, and buildings in the study area.

**Table 1.** The first-level classification of land use and cover data in the study.

| Category | Description |
| --- | --- |
| Cropland | Cropland refers to land on which crops are grown, including cultivated land, newly opened land, fallow land, swidden agriculture land, grass-field rotation land, and agricultural fruits and mulberries, mainly cultivated with crops. |
| Forest | Forest refers to forestry land where trees, shrubs, bamboo, and coastal mangrove land grow. |
| Grassland | Grassland refers to all types of grassland with a predominantly herbaceous growth and a cover of 5% or more, including scrub grassland with a largely pastoral growth and sparse grassland with a canopy density of less than 10%. |
| Waters | Waters refers to natural terrestrial waters and land for water facilities, such as lakes, ponds, reservoirs, and shallows. |
| Buildings | Buildings refers to urban and rural settlements and other industrial, mining, transportation, and other land. |
| Unused land | Unused land refers to currently unutilized land, including land that is difficult to use, such as sandy land, the Gobi, saline land, and bare land. |

The administrative boundary, highway, and railway data are from the Chinese Basic Geographic Information Center's Resource Service System (http://www.webmap.cn/, accessed on 20 September 2022). DEM data with 90 m resolution is available from the Landsat Collection 2 DEM product on the United States Geological Survey (USGS) website (https://earthexplorer.usgs.gov/, accessed on 2 May 2023). It can be used to generate slope data. Population density, GDP, soil type, and annual temperature and precipitation (2010–2020) are openly accessible from the website (http://www.resdc.cn). These data's pixel sizes are all 1 km.

Moreover, it is necessary to input multiple driving factors into the PLUS model when simulating future land use patterns. Nevertheless, this model does not support vector format. We, therefore, used the Euclidean Distance module of ArcGIS Pro software (Version 2.5, Esri, Redlands, CA, USA) to process the location point data and traffic linear data. Twenty-three remote sensing images with a spatial resolution of 250 m in 2020 were downloaded from the MOD13Q1 dataset (https://ladweb.modaps.eosdis.nasa.gov/, accessed on 10 December 2022). We used a mean value method to synthesize those images and obtain the normalized difference vegetation index (NDVI). Ultimately, there are thirteen types of drivers input into the PLUS model, including natural environment class: soil type, slope, DEM, NDVI, temperature, and precipitation; socioeconomic class: population density and GDP; and spatial accessibility: distance to rivers, governments, roads, railways, and highways. We set all data as Krasovsky_1940_Albers projection using ArcGIS Pro tools. Additionally, the ArcGIS Pro 2.5 software's resampling tool was used to unify the spatial resolution of the driver data to 30 m. The null value problem in the DEM data was addressed before input to the PLUS model. We used the conditional function in the raster calculator of the GIS tool to assign the null value to 1, and other values remain unchanged. Since the model operates automatically, it is not necessary to normalize all the drivers.

## 2.3. Methods

We constructed an ESP-PLUS framework to model multiple scenarios of LUCC considering ecological security. The framework involves four parts (Figure 2). First, multi-level ecological security patterns using MSPA and circuit theory were constructed. Second, the transition probabilities between land categories were modified by the MC model, and then we set the land demand of multiple scenarios combining the multi-ESP. Third, the outputs of the first two parts were incorporated into the PLUS model to project multiple scenarios of LUCC. Finally, to evaluate future LUCC under different scenarios, we analyzed the LUCC processes at the different levels and spatial pattern evolution.

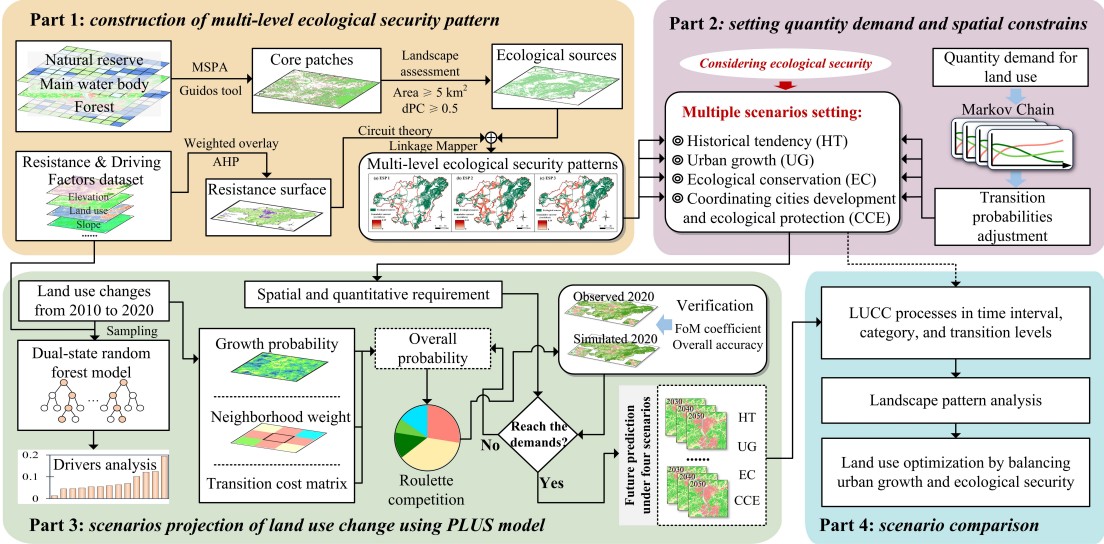

**Figure 2.** Technical framework of the study.

### 2.3.1. Land Transition Matrix

The land transition matrix can characterize conversion size among land use/cover categories, which can be deconstructed into top-down interval, category, and transition levels, as detailed in Table 2. These progressive and hierarchical analyses help scientists systematically understand the land change process. By adopting the transfer matrix and its three levels, this study explored the spatiotemporal characteristics of future LUCC patterns in the CZTMA and compared the land changes in multiple scenarios. The methodology's form is a cross-tabulation [66], whose elements on the diagonal represent the size of the unchanged areas of different land categories during a single period. Therefore, the total LUCC in a region extent is the sum of the value of other cells except the diagonal cells. If observed from a row, the non-diagonal cells represent the size that a category loses to other categories at a time interval. If observed from a column, these cells can represent the size that a category gains from other categories at a time interval. In Table 1, $B_{15}$ represents the area of cropland transition to buildings in the study area; $B_{1+}$ and $B_{+1}$ represent the areas of cropland in initial time and in the end time, respectively; $B_{1+}-B_{11}$ show the losing size of cropland; $B_{+1}-B_{11}$ show the gaining size of cropland; and $B_t$ is the total size of the study area.

**Table 2.** Land transition matrix for five categories.

| | End Time | | | | | | |
| --- | --- | --- | --- | --- | --- | --- | --- |
| | **Cropland** | **Forest** | **Grassland** | **Waters** | **Buildings** | **Total in Start** | **Loss** |
| Start time | | | | | | | |
| Cropland | $B_{11}$ | $B_{12}$ | $B_{13}$ | $B_{14}$ | $B_{15}$ | $B_{1+}$ | $B_{1+}-B_{11}$ |
| Forest | $B_{21}$ | $B_{22}$ | $B_{23}$ | $B_{24}$ | $B_{25}$ | $B_{2+}$ | $B_{2+}-B_{22}$ |
| Grassland | $B_{31}$ | $B_{32}$ | $B_{33}$ | $B_{34}$ | $B_{35}$ | $B_{3+}$ | $B_{3+}-B_{33}$ |
| Waters | $B_{41}$ | $B_{42}$ | $B_{43}$ | $B_{44}$ | $B_{45}$ | $B_{4+}$ | $B_{4+}-B_{44}$ |
| Buildings | $B_{51}$ | $B_{52}$ | $B_{53}$ | $B_{54}$ | $B_{55}$ | $B_{5+}$ | $B_{5+}-B_{55}$ |
| Total in end | $B_{+1}$ | $B_{+2}$ | $B_{+3}$ | $B_{+4}$ | $B_{+5}$ | $B_t$ | |
| Gain | $B_{+1}-B_{11}$ | $B_{+2}-B_{22}$ | $B_{+3}-B_{33}$ | $B_{+4}-B_{44}$ | $B_{+5}-B_{55}$ | | |

### 2.3.2. Identifying Ecological Sources

We applied the morphological spatial pattern analysis (MSPA) to conduct a comprehensive landscape analysis and then extracted ecological sources through connectivity assessment. MSPA is an identifying and segmenting method of raster image processing by erosion, expansion, and open/close arithmetic, which can effectively determine the landscape type and structure [67]. The forest and waters have good ecological functions and less disturbance by human activities, which are ideal spaces for species to survive [68,69]. Buildings and cropland are greatly affected by human activities [70,71]. Moreover, the grassland's ecological quality in the study area is poor compared with the forest, and the grassland lacks a better environment for species to forage and rest. Hence, based on the previous works, we selected forest and waters in 2020 as foreground pixels and other land use and cover categories in 2020 as background pixels. Using Guidos-Toolbox software (Version 3.2, European Commission, Joint Research Centre (JRC), Ispra (VA), Italy, https://forest.jrc.ec.europa.eu/en/activities/lpa/, accessed on 12 February 2023) [72,73], the shape, connectivity, and spatial arrangement detection of binary images in 8-bit tiff format were performed using the eight-neighborhood approach. Next, the images were segmented into seven landscape types: core, islet, perforation, edge, loop, bridging, and branch.

Ecological sources are areas with sufficient material exchange and energy movement, and their precise identification is critical for constructing ecological security patterns [74]. Specifically, this study considered the size and connectivity of patches to select ecological sources, which avoids subjectively defining core ecological sources as large patches. We inputted the core area recognized by MSPA into the Conefor Sensinode (Version 2.6, Uni-

versity of Lleida, Lleida, Spain) and then computed the connectivity. The Conefor tool and its manual are accessible on the website (http://www.coneforsensinode.html, accessed on 17 May 2023) [75]. Equations (1) and (2) give the formulas to calculate the probability of connectivity (PC) and the delta of PC (ΔPC) for each patch [67]. The PC and ΔPC can evaluate regional landscape connectivity and the importance of patches to landscape connectivity, respectively. Finally, we regarded the patches with an area larger than five square kilometers and a ΔPC value greater than 0.3 as ecological sources.

$$PC = \frac{\sum_{i=1}^{n}\left(\sum_{j=1}^{n} a_i \times a_j \times P_{ij}^*\right)}{S_t^2} \tag{1}$$

$$\Delta PC = \frac{PC - PC_i}{PC} \times 100\% \tag{2}$$

where $S_t$ is the total source area; $n$ represents the number of source patches; and $a_i$ and $a_j$ are the patches $i$ and $j$ sizes, respectively. $P_{ij}$ shows the possible connectivity index between patches $i$ and $j$, and its range is 0–1. A larger PC represents better connectivity among patches; $PC_i$ is the value of the overall connectivity after removing patch $i$. A more excellent ΔPC value represents a higher importance of the patch to sustain landscape connectivity.

### 2.3.3. Setting Resistance Surface

The resistance surface is the spatial distribution patterns of ecological process resistance. Firstly, a resistance surface factor system was established, combining previous studies and the study area [76]. We chose slope, elevation, land use category, MSPA landscape type, and distance to buildings as resistance factors. Then, the analytic hierarchy process (AHP) was used to determine the weights of each factor (Table 3).

**Table 3.** Design and weight of resistance surface factors.

| Resistance Factor | Weight | Resistance Value | | | | |
|---|---|---|---|---|---|---|
| | | 1 | 3 | 5 | 7 | 9 |
| Slope | 0.17 | <3° | 3°–8° | 8°–15° | 15°–25° | >25° |
| Elevation | 0.07 | <50 m | 50–150 m | 150–250 m | 250–350 m | >350 m |
| Land use category | 0.37 | Forest | Grassland | Cropland | Waters | Buildings |
| MSPA landscape type | 0.17 | Core | Loop, Bridge | Islet, Branch | Perforation, Edge | Background |
| Distance to buildings | 0.22 | >1500 m | 1000–1500 m | 500–1000 m | 200–500 m | <200 m |

### 2.3.4. Extraction of Ecological Corridors

Corridors show pathways communicate ecological sources. They are crucial structural areas for improving landscape connectivity in ecological restoration programs to decrease the probability of interception and segmentation for ecological processes [77]. Circuit theory abstracts patches as nodes and resistances [74]. It can identify ecological corridors using thoughts of random wandering characteristics of charges and a minimum cumulative resistance (MCR), as detailed in Equation (3).

$$MCR = f_{min}\sum_{j=y}^{i=x} D_{ij} \times V_i \tag{3}$$

where MCR is the minimum cumulative resistance value from an ecological source to grid unit; $f$ is a function that indicates the positive correlation between minimum cumulative resistance and ecological processes; $D_{ij}$ is the spatial distance from source $j$ to grid $i$; and $V_i$ is the resistance value in grid $i$.

We identified the ecological corridors of the CZTMA using the GIS tool Linkage Mapper (http://www.circuitscape.org/linkagemapper, accessed on 18 May 2023). The corridor areas were determined according to the threshold of cumulative resistance, and the areas where the cumulative resistance exceeded the threshold were excluded. The detailed

identification process is as follows. First, the neighboring source areas were identified using cost allocation and Euclidean allocation functions in ArcGIS software (Version 10.2, Esri, Redlands, CA, USA). Based on the source polygons and the resistance raster, the raster file was created to assign grid cells to the nearest Euclidean space of a source area. Second, a network of source areas was built via adjacency and distance data. This network includes line data connecting two patches, resulting in a stick map. The attribute of the stick map represents the Euclidean distance between patches. Third, we performed a cost-weighted distance computation for each source area based on the source polygons, resistance raster, and linked table from step one. Finally, the minimum cumulative cost distance between a pair of source areas was acquired, and then the shortest path was identified. Such an operation requires $n(n-1)/2$ times where $n$ shows the number of sources until all source areas are calculated, forming all corridors among the ecological source. An ecological corridor is the optimal corridor with the lowest cumulative cost consumption and the shortest distance, and these corridors cannot be a straight line but rather an irregular curve, as shown in Figure 3.

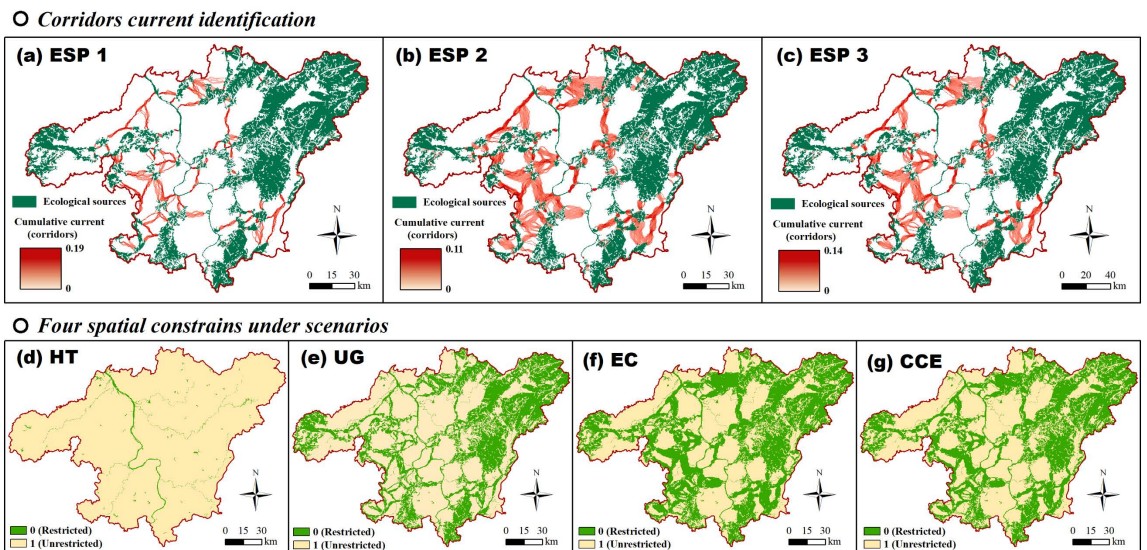

**Figure 3.** Setting of ecological constraints under the four scenarios. Note: identification of three levels of ESPs (**a**–**c**); ecological constraint designations for different scenarios (**d**–**g**); HT: historical tendency; UG: urban growth; EC: ecological conservation; CCE: coordinating city development and ecological protection.

2.3.5. Different Levels of ESPs and Multi-Scenario Settings

Different ESP levels were structured according to the extent of ecological corridors formed by various cumulative resistance thresholds. Studies have found that the minimum width required to protect birds is 200 m, the minimum width required to form a diverse landscape structure is 600 m, and the minimum width required to foster a close natural habitat is 1200 m [78,79]. According to previous research, ecological corridors with cumulative resistance thresholds of 1000, 2400, and 3200 were designed, corresponding to ESP1, ESP3, and ESP2 [43]. Also, Figure 3 shows the four spatial constraints under different scenarios.

Changes in demand for future development impact land use evolution. The land use prediction in a metropolitan area under multiple scenarios is designed to provide decision-makers with various decision-making perspectives. These perspectives can help planners scientifically judge the spatial patterns and transition direction among land use types, which are significant to the harmony of human–land relationships and sustainable socioeconomic development. Therefore, we have designed the four scenarios, as shown in Table 4.

**Table 4.** Four scenario designs regarding land change modeling.

| Scenario | Description | Parameter |
|---|---|---|
| Historical tendency (HT) | This scenario continues the evolutionary trend of the different land categories over the historical period (2010–2020) and is the reference for the parameters setting of the other scenarios. The spatial constraint for the HT scenario consists of the key water bodies, corresponding to Figure 3d. | This scenario maintains the probabilities of transitions among land use categories calculated from Markov chains between 2000 and 2020. We extrapolated the land use demands in 2030, 2040, and 2050, starting with 2020. |
| Urban growth (UG) | This scenario mainly focuses on urban expansion to ensure regional economic advances. The loss of forest, grassland, and cropland accelerates. It also added ESP1 spatial constraint to the HT scenario, corresponding to Figure 3e. | The probability of a transition from cropland, forest, and grassland to buildings increases by 20%; the probability of a transition from buildings to other categories decreases by 30%, while the probability of a transition from buildings to cropland stays. |
| Ecological conservation (EC) | In the context of emphasizing ecological restoration and protection, ecological land is prioritized for protection to reduce its shrinkage rate. The ESP2 spatial constraint was added to the EC scenario, corresponding to Figure 3f. | We set the probability of a transition from other categories to buildings decreased by 30%, and the probability of a transition from buildings to others increased by 20% (except for the transition to cropland) |
| Coordinating city development and ecological protection (CCE) | The CZTMA is in a critical period of economic recovery and industrial upgrading. Appropriate growth of urban areas remains a realistic need for the socioeconomic development of the region, but in accordance with the requirements of ecological civilization. The ESP3 spatial constraint was added to the scenario, corresponding to Figure 3g. | The probability of a transition from forest and cropland to buildings reduced by 10%; the probability of a transition from waters and grassland to buildings decreased by 20%; the probability of a transition from buildings to forest reduced by 20%; the probability of a transition from buildings to waters and grassland reduced by 10%. |

### 2.3.6. LUCC Scenario Simulation

The PLUS model is comprised of a rule mining framework that utilizes the land expansion analysis method (LEAS) and a cellular automaton (CA) model that is based on multi-type random patch seeds (CARS) [29]. Initially, the LEAS module is responsible for extracting the expansion pattern of each land category from the land use maps at two time points. The subsequent analysis elucidates the variations in the impact of individual driving variables on LUCC, as determined by implementing the random forest classification (RFC) algorithm. Next, the module can understand the land transition rules, calculate each land category's probabilities, and produce their spatial distribution map. Equation (4) shows the formula to compute the probability of land category $k$ on pix $i$ [29].

$$P_{i,\,k}^{d}(x) = \frac{\sum_{n=1}^{M} I = \{h_n(x) = d\}}{M} \tag{4}$$

where $d$ is a binary variable that takes 0 or 1; if $d$ equals 1, it shows other land categories converted to category $k$; if $d$ equals 0, it shows the other transition occurs. The vector $x$ is comprised of many drivers, where each driver represents a certain component; $h_n(x)$ represents the predicted category of the $n$-th decision tree's vector $x$; and $M$ shows the total number of decision trees in the analysis. The indicator function $i$ is used to denote the decision tree, indicating if a particular driver is present or not.

After the steps in the above module are completed, the CARS module requires the input of development probabilities, neighborhood weights, and transition costs for all categories. Also, we can import a constraint factor into this module. According to Equation (5), the neighborhood weights for cropland, forest, grassland, waters, and buildings were set in Table 5, respectively, according to the percentage of area in category change. The

model incorporates a combination of stochastic seed generation and threshold-decreasing mechanisms to simulate the generation of patches. Each patch is ultimately represented as a land category, which is determined by the overall probability of the land use category as described in Equation (6) [29]. The feedback mechanisms of macro-demand and local competition influence the specific outcome of the model.

$$Weight_i = \frac{Gain_i}{\sum_{i=1}^{5} Gain_i} \tag{5}$$

$$OP_{i,k}^{d=1,\,t} = P_{i,k}^{d=1} \times \Omega_{i,k}^{t} \times D_k^t \tag{6}$$

where $Weight_i$ is the weight of land category $i$; $Gain_i$ denotes the expansion area of category $i$ between 2010 and 2020; $i$ represents one of the five land categories in the study area; $OP_{i,k}^{d=1,\,t}$ is the overall probability of land category $k$ on pix $i$; $P_{i,k}^{d=1}$ is the transition probability of land use category $k$ on pix $i$; $\Omega_{i,k}^{t}$ denotes the neighborhood effect of land category $k$ on pix $i$; $D_k^t$ shows the influence of future demand for land category $k$ and also represents the adaptive drive coefficient, whose magnitude is decided by the deviation between the iterated quantity and the macro demand for category $k$ at time $t$.

**Table 5.** Neighborhood weights for land category.

| Land Category | Cropland | Forest | Grassland | Waters | Buildings |
|:---:|:---:|:---:|:---:|:---:|:---:|
| Weight | 0.251 | 0.257 | 0.008 | 0.033 | 0.451 |

The overall accuracy (OA) index was used to check the PLUS model's simulation performance. The OA is the proportion of pixels consistent between simulated and observed land use maps in 2020. We also used the figure of merit (FoM) coefficient to determine how accurate the PLUS simulation was by comparing the change in the simulation to the change seen from 2010 to 2020 [80]. The final OA and FoM values for the simulated land use map are 92.5% and 0.22, respectively. According to previous investigations, the precision results were acceptable. The FoM is computed in the following manner:

$$\text{FoM} = \frac{Hits}{Misses + Hits + False\ alarms + Wrong\ hits} \tag{7}$$

where *Misses* is the pixels that simulate as change but are observed as unchanged; *Hits* is the pixels that are both simulated and observed as change; *False alarms* is the pixels that are simulated as unchanged but observed as change; and *Wrong hits* is the pixels that are both simulated and observed as change but simulated as an error category.

### 2.3.7. Landscape Pattern Metrics

Landscape metrics can quantitatively express the characteristics of patch, shape, and clustering in land use/cover patterns [81]. We selected eight indices, including the number of patches, mean patch size, edge density, fractional dimension, contagion, aggregation index, Shannon's diversity index (SHDI), and Shannon's evenness index (SHEI). Land use maps were converted to raster format using ArcGIS Pro software. Then, we calculated these indices based on Fragstats software (Version 4.2, Portland, OR, USA).

## 3. Results

### 3.1. Ecological Security Pattern in the CZTMA

Figure 4 shows the different landscape types in the study area using the MSPA method. According to the data presented in Figure 4, the cumulative area of the seven distinct landscape types amounts to 11,639.2 km$^2$ and 61.5% of the overall research area. The core area has the largest expanse, measuring 7816.4 km$^2$, while the bridge area represents the smallest region, spanning a mere 43.8 km$^2$. The primary region is dispersed across the

eastern and southern sectors of the CZTMA, predominantly inside the Luoxiao Mountains (Figure 4a). The center and western regions have fewer patches and a more fragmented core area, which hinders the efficient cycling of materials and energy conversion in ecological processes. The perforation and edge areas serve as protective buffers for the core region, acting as transitional zones between the core area and other landscape types. These areas account for 2.1% and 13.2% of the study extent. It reveals that the core patches in the CZTMA have stability and can resist the impacts produced by the disturbances of external influences.

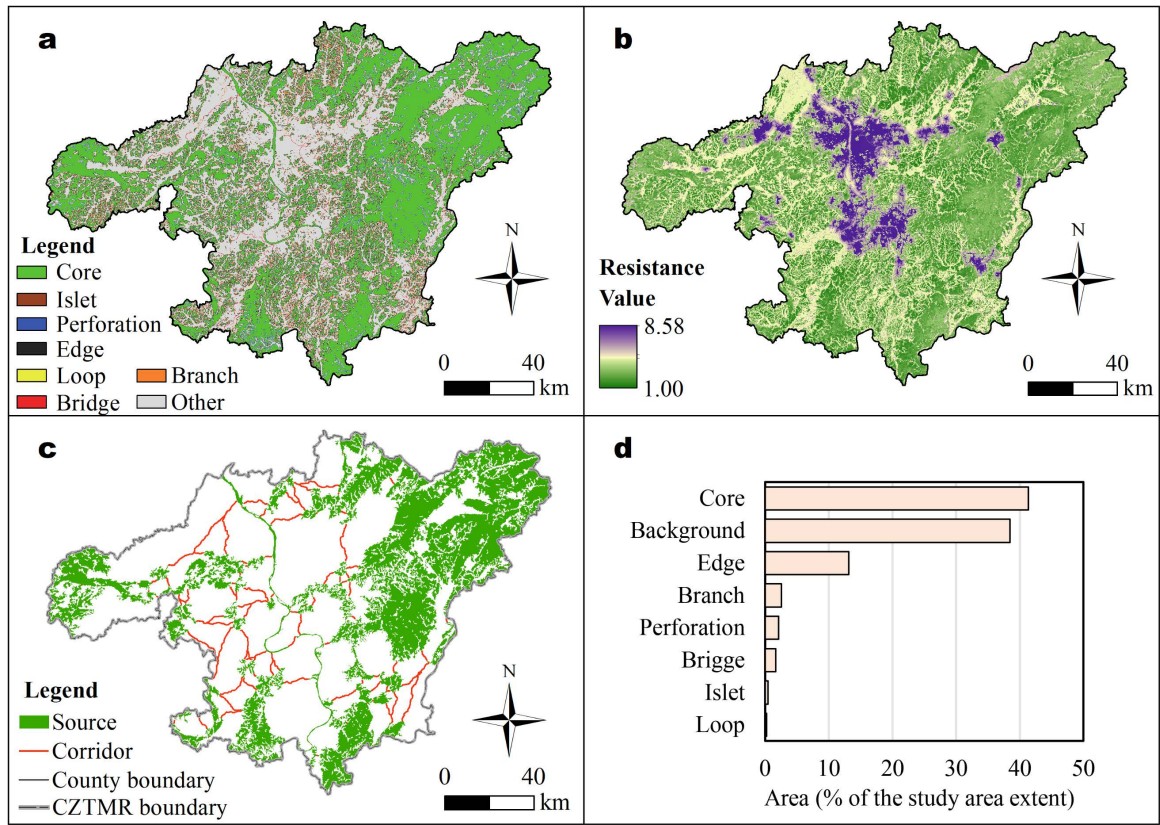

**Figure 4.** Identification of ecological sources and corridors in the CZTMA. Note: (**a**) landscape distribution under the MSPA method; (**b**) resistance surface; (**c**) ecological security pattern; (**d**) landscape structure.

Determining the weights of the five factors using the AHP method, we obtain a comprehensive resistance surface of the CZTMA by weighted overlay (Figure 4b). The northern and central parts of the study area have higher resistance values due to the influence of cropland and buildings in these areas. The resistance distribution is high in the center and low in the surroundings, which could affect the exchange of ecological information and species migration between the eastern and western core areas. Finally, 46 ecological corridors were estimated, of which the length of 26 corridors was greater than 15 km and had a maximum length of about 126 km (Figure 4c).

### 3.2. Future Prediction of LUCC under Multi-Scenario

We inputted the parameters of the four modeling scenarios into the CARS module. This study forecasted the LUCC dynamics for the next 30 years by taking 2020 as the base year, as shown in Figure 5 and Table 6.

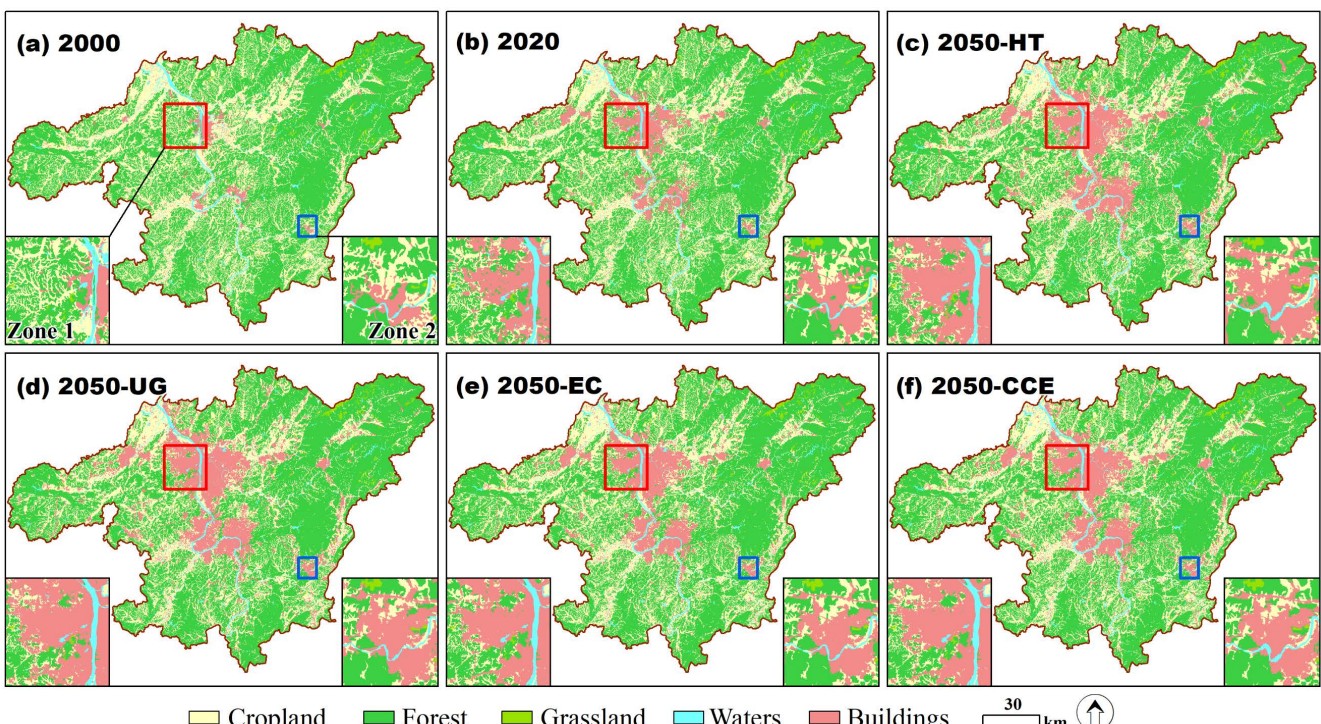

**Figure 5.** Historical land use/cover maps (**a**,**b**) and simulated maps (**c**–**f**) in 2050 under the four scenarios in the CZTMA. HT: historical tendency; UG: urban growth; EC: ecological conservation; CCE: coordinating city development and ecological protection.

**Table 6.** Change dynamics in land categories' area during 2000–2050 (Unit: km²).

| Scenario | Year | Cropland | Forest | Grassland | Waters | Buildings |
|---|---|---|---|---|---|---|
| Actual observation | 2000 | 6210.5 | 11,593.7 | 172.2 | 412.0 | 526.4 |
| | 2010 | 5919.2 | 11,383.4 | 173.4 | 429.7 | 1009.3 |
| | 2020 | 5681.6 | 11,208.5 | 171.2 | 430.8 | 1422.9 |
| Historical tendency | 2030 | 5467.8 | 11,014.3 | 168.4 | 430.2 | 1801.1 |
| (HT) | 2040 | 5283.8 | 10,853.1 | 166.3 | 431.1 | 2147.4 |
| | 2050 | 5122.2 | 10,723.9 | 164.2 | 432.1 | 2439.4 |
| Urban growth | 2030 | 5421.5 | 10,963.2 | 168.1 | 427.2 | 1901.8 |
| (UG) | 2040 | 5203.8 | 10,775.8 | 166.2 | 426.8 | 2309.1 |
| | 2050 | 5010.3 | 10,570.2 | 163.8 | 424.6 | 2712.7 |
| Ecological conservation | 2030 | 5537.9 | 11,083.9 | 168.7 | 419.5 | 1671.7 |
| (EC) | 2040 | 5582.4 | 11,015.8 | 167.6 | 415.6 | 1700.3 |
| | 2050 | 5309.0 | 10,926.3 | 166.0 | 407.2 | 2073.2 |
| Coordinating city development | 2030 | 5491.3 | 11,029.1 | 168.5 | 431.6 | 1761.2 |
| and ecological protection (CCE) | 2040 | 5332.1 | 10,906.3 | 167.2 | 425.6 | 2050.6 |
| | 2050 | 5187.5 | 10,762.2 | 165.3 | 437.4 | 2329.4 |

In 2020, the study area was dominated by forest, cropland, and buildings, with 5681.6 km², 11,208.5 km², and 1422.9 km², respectively. Figure 5 shows that the land use structure's characteristics will remain stable under all the scenarios during 2020–2050. Land-use categories change to concentrate on cropland, forest, and buildings. The expansion of buildings is evident. Each land use type remains consistent in spatial distribution, with cropland mainly distributed in the plains with flat terrain and well-developed water systems. Forest is predominantly spread in hilly and mountainous locations with higher topography in the west and south. Buildings are mainly clustered on both sides of the rivers, that is, along the Xiangjiang River and its tributaries, the Liuyang River and the Weishui River. The extension of buildings is predicated upon the initial spatial allocation and further

extends in a linear fashion adjacent to the river. Urban growth occurs mainly in Changsha County, Wangcheng District, Yuhu District, Ningxiang City, and Tianyuan District by occupying cropland and forests that have ecological functions. If land urbanization is not restricted, it will threaten the ecological security of the metropolitan area.

There are differences in the dynamic trends of land categories during 2020–2050 (Table 6). The buildings' area has increased, while cropland, forest, and grassland have all shrunk, and waters is unstable. Under the HT scenario, the buildings' area will be 2439.4 km$^2$ by 2050, which is an increase of 71% compared to 2020. The buildings' area in 2050 is 2712.7 km$^2$, which is an increase of 90% compared to 2020 under the UG scenario. In the EC and CCE scenario, the buildings' areas in 2050 are 2073.2 km$^2$ and 2329.4 km$^2$, respectively. Among land types, the building expansion is the most significant. In the context of urbanization, without strong policy intervention measures, city construction areas will continue to sprawl to meet the needs of urban population growth and economic development for land carriers. Urban expansion will mainly sacrifice forest and cropland. The future directions of forest and cropland inherit the feature in the historical period 2000–2020, showing a net decrease. During the 2020–2050 HT scenario, the area of cropland and forest decreased by 559.4 km$^2$ and 484.6 km$^2$, respectively. Compared to the HT scenario, the cropland, forest, and grassland change rates were reduced in the EC scenario. Furthermore, the EC scenario may effectively reduce the environmental pressure brought on by built-up areas engulfing agriculture and forests under the influence of ecological network restrictions, which gives decision-makers a foundation for ensuring the security of the regional ecosystem. Nonetheless, it is challenging to prevent a rise in the size of buildings when local governments are tasked with establishing goals for superior economic growth. Forest and cropland will still be compressed to some extent in the future, although the EC scenario can effectively control the rate of ecological land reduction.

### 3.3. Scenarios Comparison and Analysis

#### 3.3.1. Comparison Based on the LUCC Processes

Figure 6 shows the LUCC processes and their impact on the ecological area under future scenarios during 2020–2050. From the time interval level (Figure 6a), the land use's overall change under the four future scenarios is UG > HT > CCE > EC. The change size in the UG scenario reached 2938.95 km$^2$ (15.55%), with the most drastic land dynamics. It demonstrates that the priority given to enhancing urban and rural construction in the future possibly results in a speedy rate of land change. Compared with the UG scenario, the EC scenario's land change area is 6.79% of the study area, reducing the scale by half, which implies that appropriate ecological network protection measures in the future can effectively curb the size of land use change in the metropolitan area.

There were differences in the category's gain and loss under different scenarios (Figure 6b). Regarding category loss, the forest is the largest (3.53%–6.22%), followed by cropland (3.41%–4.53%). Additionally, the buildings' gain is huge (3.72%–7.14%) despite the small proportion of buildings within the study extent. From the perspective of scenarios, the areas of forest and cropland loss under the UG scenario are 1177.47 km$^2$ (6.23%) and 979.12 km$^2$ (5.18%), respectively, indicating a significant deforestation process. Under the EC scenario, the shrinkage in forest and cropland can be mitigated. Regarding the transition size to buildings (Figure 6d), the UG scenario has an enormous transition size, and the EC scenario has the smallest. Between 2000 and 2050, the transition area from forest to buildings is 381.78–893.97 km$^2$ (2.02%–4.73%), with the most significant UG scenario. Throughout the four scenarios, it is clear that the overall characteristic of the LUCC processes is that the CCE scenario falls between the HT scenario and the UG scenarios, and the UG scenario is larger than the historical trend scenario.

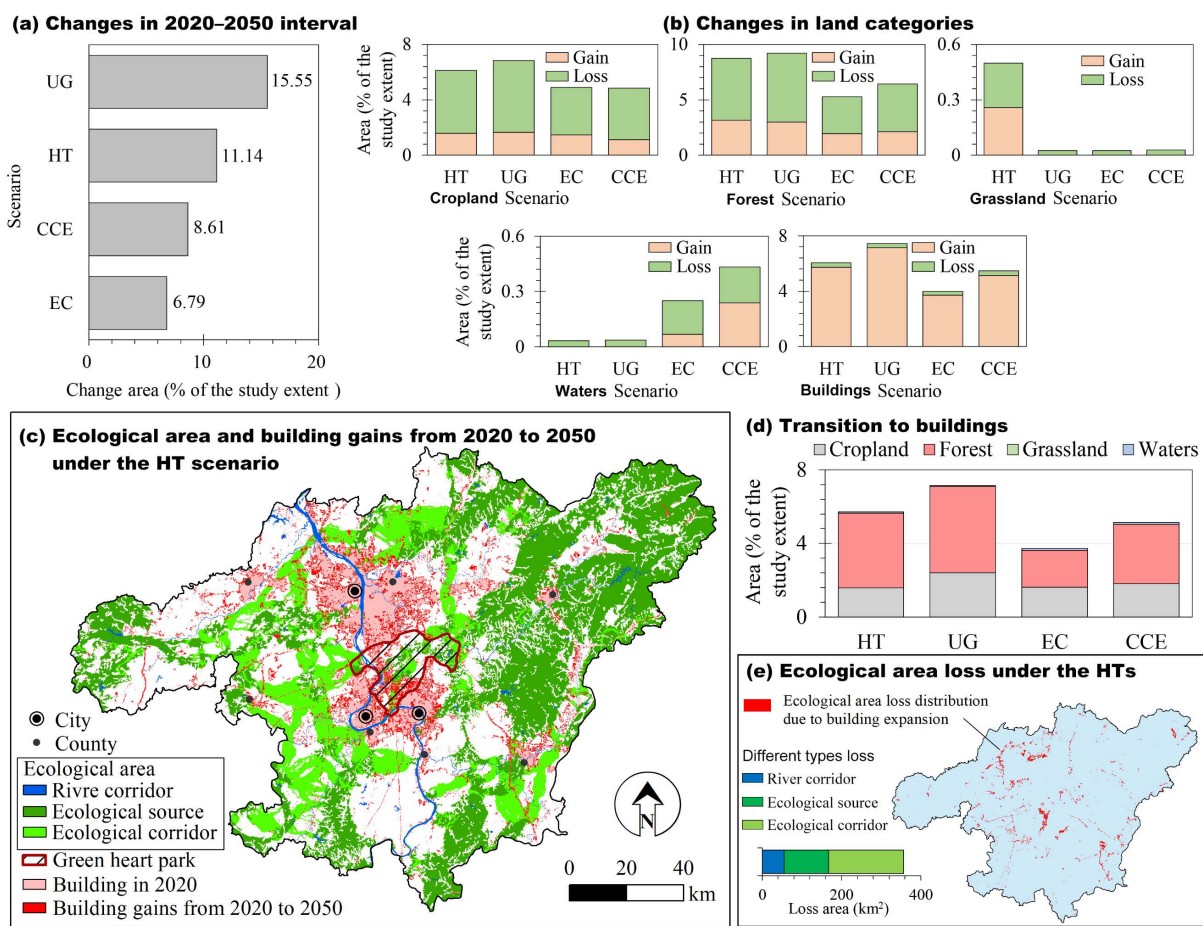

**Figure 6.** Scenario comparison in the land change process during 2020–2050. Note: different scenarios' overall changes at interval level (**a**); scenario differences at category level (**b**); ecological area and buildings gain from 2020–2050 under the HT scenario (**c**); other land categories' transition to buildings (**d**); ecological area loss distribution due to buildings growth under the HT scenario (**e**); HT: historical tendency; UG: urban growth; EC: ecological conservation; CCE: coordinating city development and ecological protection.

We overlaid the building expansion under the HT scenario and the ESP3 level ecological area. Then, we identified the loss distribution of different ecological areas caused by building growth (Figure 6c,e). As shown in Figure 6c, building expansion mainly takes buildings in 2020 as the source of growth, significantly encroaching on critical ecological land that maintains ecological security. Also, the increase of buildings has occurred in the green heart area of the CZTMA. Therefore, in the future, the development of green heart parks should pay attention to protecting the core ecological patches within the region to prevent "constructive" damage to the surrounding ecological environment. In the context of the HT scenario, it is seen that the ecological regions occupied by buildings are primarily concentrated in the central region of the CZTMA and exhibit proximity to the metropolitan center. Hence, using the ESP as a restricted conversion area can help optimize future urban land growth, protect core ecological areas, and ensure crucial ecological processes in socioeconomic development. Figure 6e shows that the main types of ecological loss from 2020 to 2050 will be ecological corridors (188.79 km$^2$), followed by ecological source areas and river corridors.

### 3.3.2. Comparison Based on Pattern Metrics

Figure 7 shows the dynamic situations of eight pattern indices from 2000 to 2050 in the different scenarios. As shown in Figure 7, the patch numbers in the future scenarios

are greater than in the historical period and show an upward trend. In contrast, the future mean patch sizes are smaller than the historical period and decrease. Compared to the other scenarios, the UG scenario has the most enormous patch numbers and the smallest mean patch sizes. In future scenarios, edge density and fractional dimensions showed an increase and decrease, respectively, but the change in fractional dimensions was small. The values of contagion and aggregation keep getting smaller. The trend with the mean patch size characteristics could reflect the dispersion and fragmentation of patches in the CZTMA. It results in spatial isolation of ecological patches. The SHDI showed an increasing trend, especially in the HT scenario. Also, the SHEI characterizes the landscape equilibrium development and shows a continuous increase.

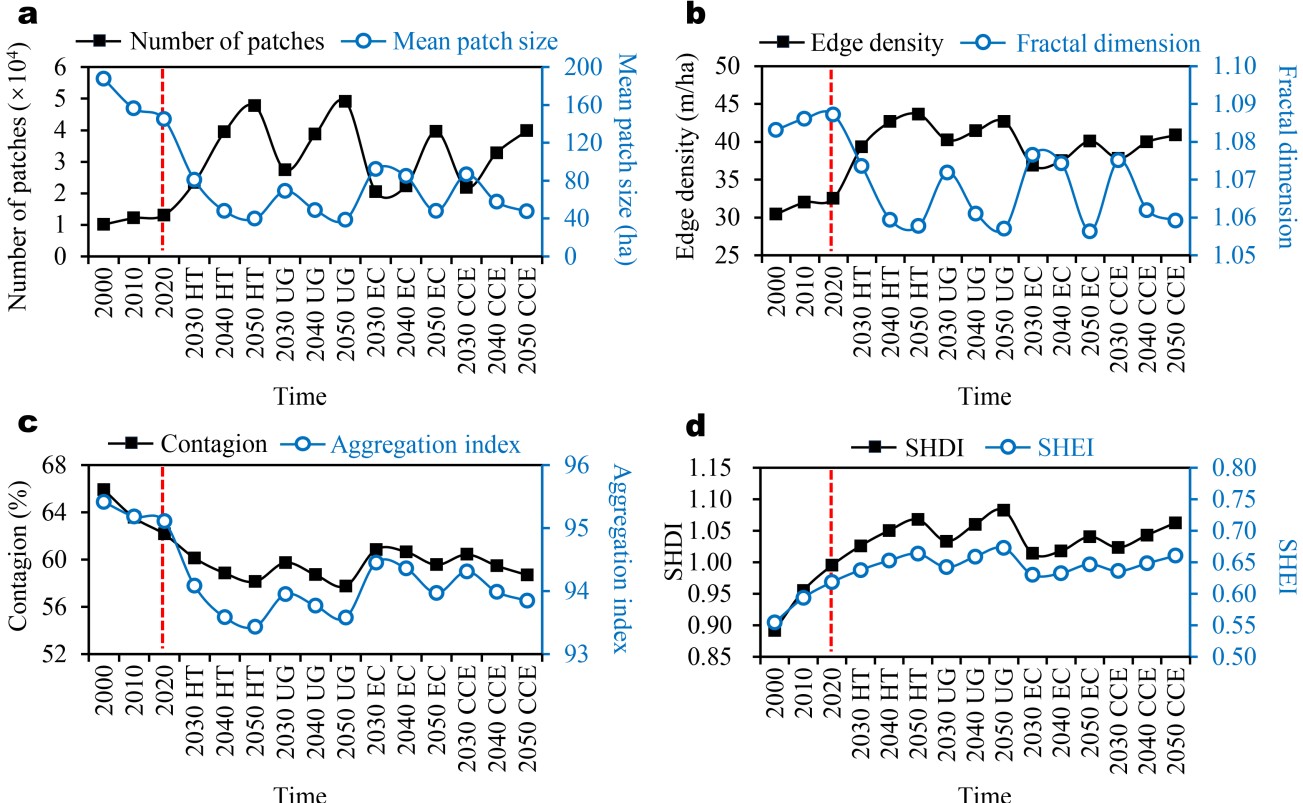

**Figure 7.** Landscape pattern analysis in the CZTMA. Note: number of patches and mean patch size (**a**); edge density and fractal dimension (**b**); contagion and aggregation index (**c**); SHDI and SHEI (**d**); HT: historical tendency; UG: urban growth; EC: ecological conservation; CCE: coordinating city development and ecological protection. The left side of the red dashed line shows the historical period, and the right side shows different scenarios in the future.

### 3.4. Driving Forces Analysis Using the PLUS Model

Thirteen geographical variables were chosen as influential aspects derived from the natural environment, spatial proximity, population, and economics. The LEAS module utilized the land use maps from 2010 and 2020 and spatial drivers, as input data. We set random sampling by category proportion; the sampling rate is 0.01, the number of decision trees is 20, and the number of training features of random forest is 13. Figure 8 shows the degree of contribution of each driving factor to the change of land categories in the study area.

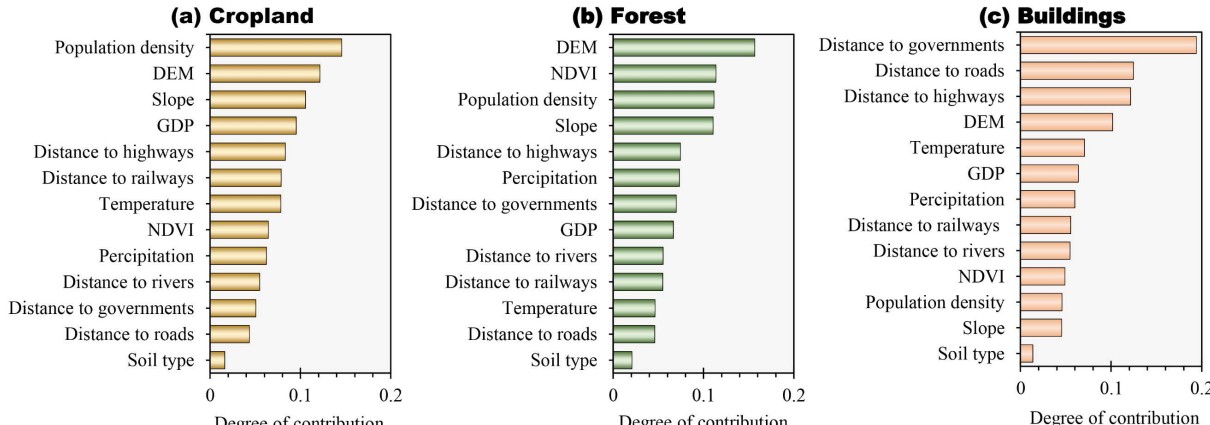

**Figure 8.** Contribution of spatial variables to the dynamic changes of (**a**) cropland, (**b**) forest, and (**c**) buildings in the CZTMA.

As shown in Figure 8a, the major factors influencing the change in cropland are population density, elevation, and slope. Cropland significantly contributes to the growth of buildings in Figure 6b. On the one hand, population density impacts cropland's dynamics because population density determines the demand for residential, transportation, and productive lands. The population constitutes the primary entity of urban growth, and, to a certain degree, it indicates a city's liveliness. The population density increased from 656 people/km$^2$ in 2000 to 899 people/km$^2$ in 2020 in the CZTMA, and the population was concentrated in the city. The rising population density in urban areas means expanding buildings, thereby modifying the spatial pattern of forest and cropland around the cities. Thus, the reduction of forest and cropland is likely influenced by housing and infrastructure development. On the other hand, cropland serves as an essential space for local food production. Under the influence of China's cropland occupation and supplementation policy, the loss of cropland due to urban sprawl can be accompanied by increasing cropland in the periphery away from the cities.

The main driver affecting the forest is elevation, followed by NDVI, population density, and slope (Figure 8b). The distances to government, roads, and railroads impact buildings more in Figure 8c. In addition, elevation as a natural factor contributes relatively more than other natural factors. Meanwhile, administrative and transportation location accessibility significantly impacts buildings because urban land, rural settlements, and industrial land usually spread along both sides of the road. The development of the CZTMA is driven by the internal desire for ongoing urban growth, which aligns with the plan for the rise of central China. The growth needs to comply with the requirements of requisition-compensation balance of cropland and ecological civilization construction in China. Therefore, analyzing the driving factors for the shrinkage of cropland and forests can similarly reflect the reasons for the building expansion.

## 4. Discussion

### 4.1. The ESP-PLUS Framework to Balance Urban Growth and Ecological Protection

This ESP-PLUS framework can quantitatively and spatially restrict the future land use evolution, which provides a technical approach for balancing ecological protection and urban development. Many studies have set the land demand under different scenarios using MC, SD, MOP, and transition cost methods to optimize future land use patterns [12,19,82]. Nonetheless, they could ignore the spatial constraints on the land use modeling process. As a result, these investigations to achieve spatial optimization of land use are challenging despite realizing it in quantity. For example, the eastern part of a city is highly productive cropland, while the northern part is unproductive grassland. If only the quantitative constraints on urban areas were applied, preventing urban sprawl from encroaching on these croplands would be difficult. Suppose we spatially and quantitatively

restrict the city's eastern part to experience transformation. In that case, the city will expand northwards or in some other direction instead of eastwards in the simulated results. When Zhang et al. [41] predicted LUCC in the Yangtze River Delta (YRD) region, they considered the ecological sources derived from ecosystem services analysis to be a spatial constraint. Liu et al. [83] extracted key ecological areas as limiting factors to optimize land use in Jinan City according to high ecosystem service value (ESV) principles. Also, scientists input natural reserves and ecological red lines as restriction factors in the LUCC simulation model [33,40]. Although previous studies considered ecological protection, few involved underlying corridors between ecological zones when forecasting LUCC. These studies would have been more interesting if they had synthesized the ecological security pattern containing pathways to connect core patches. In addition, researchers have constructed future ecological security patterns based on predictive land use and cover maps [84,85]. Their modeling ideas are helpful but differ from our study.

The validated results from this study's simulation have confirmed the ESP-PLUS framework's efficacy in ecological land conservation. The forest area will decline in the future, similar to the trend from 2000 to 2020 in Table 6. The apparent shrinkage of the forest area was curbed under the EC and CCE scenarios (Figure 6). Furthermore, landscape fragmentation was comparatively smaller in the EC and CCE scenarios than in the remaining two (Figure 7). The results are similar to some studies that have employed ecological source lands as spatial constraints [37,84]. In contrast to traditional ecological limits such as nature reserves, national parks, rivers, and forests, our ecological protection scenarios incorporate corridors and their connectivity qualities, showing a more comprehensive representation of an ecosystem. This study broadens the research perspective of a multi-scenario simulation of land use to some extent.

### *4.2. Relevant Development Suggestions*

Urban expansion and dramatic human activities significantly alter the structure and function of land cover [7]. Under the principle of maximizing economic interests, the structural integrity of ecosystems is neglected in urban development, resulting in the excessive occupation of ecological land by urban land, which leads to the degradation of ecosystem functions. Understanding LUCC is crucial in protecting the regional ecological security pattern [52]. Forests, water areas, and grasslands with high coverage have a good ecosystem service function [86,87]. The building area in the CZTMA increased from 526.4 km$^2$ in 2000 to 1422.9 km$^2$ in 2020 (Table 6). Under various scenarios, the predicted area will reach 2073.2–2712.7 km$^2$ in 2050. Such expansion sprawl may cause irreversible environmental destruction if this region's key ecological zones (i.e., sources and corridors) are not protected. There are 19 county-level governments in the CZTMA. The county level is the basic administrative unit in China. The life of planning depends on the implementation [67,76]. A problem is that every local government has independent plans and intentions, so they need a unified understanding of the overall planning. It divides the whole ecosystem and regional development to a certain extent and might indirectly cause the loss of socioeconomic and ecological benefits. Therefore, we suggested that establishing a provincial-level senior leadership coordination mechanism strengthens the coordination and guidance for the development of the CZTMA, which improves the ability to resolve significant challenges in a coordinated manner and minimizes unhealthy competition among cities.

In the plan for the metropolitan area, a central park will be built in the middle of the three cities and along the Xiangjiang River [54]. The park, also known as the "Green Heart", covers an area of approximately 523 km$^2$. It will provide the surrounding residents with eco-tourism vacations and rural leisure services. At the same time, it will facilitate the development of future eco-industries in the CZTMA, which promotes natural environment conservation. Additionally, we found that the ecological patches at the central park are one of the vital pathways to connect the ecological sources on both sides of the study area (Figure 4c). It implies that the park provides essential residence for species that need

long-distance migration and wildlife habitats, which can help sustain regional biodiversity. Habitat fragmentation caused by anthropogenic activities has affected the movement and dispersal of wildlife, eventually leading to an increased probability of species extinction [52]. Therefore, planners and managers must give precedence to the conservation and restoration of ecological areas in the core region of the study area while also taking measures to avoid the depletion of ecological corridors dispersed throughout agricultural land.

Firstly, the local government should firmly grasp the "overall situation", adhere to the basic guiding principles of coordinated regional development and green ecological priority, and optimize the mechanism of coordinated diagnosis, treatment, and restoration of the ecological environment. Secondly, the city's planning department should consider the protection and development of the ecological green heart within the metropolitan area and, on the one hand, strengthen the ecological restoration of the fragile areas of the green heart and strictly control the ecological red line area. On the other hand, it is necessary to optimize and revitalize the "stock" of buildings and improve the level of intensive and economical use. Finally, it is necessary to pay attention to the importance of waters and wetlands in maintaining the diversity of ecosystems in the region as well as regulating the hydrological process, setting up a blue line for the protection of key waters, and strengthening the management and protection of the shorelines of Xiangjiang River and other lakes to ensure the safety of water resources.

Liuyang City dominates the eastern part of the CZTMA. The region is at a relatively high altitude and has a wide distribution and good connectivity of ecological sources, hence the relatively few corridors. The local government has recently introduced many papermaking, furniture manufacturing, and wood processing enterprises. They have become an essential part of supporting the local economy. However, these companies have also brought tremendous pressure on local environmental protection. For example, papermaking generates sewage that contains large amounts of poisonous and hard-treated chlorinated organic matter, which is strongly toxic to wildlife and humans. Moreover, papermaking requires much wood, inevitably causing extensive deforestation. The residents and the government should strengthen the regulation of effluent treatment and increase the penalties for improperly operated enterprises. The environmental administration departments should enhance their dynamic observation of ecological sources and corridors using remote sensing and GIS technology to prevent critical ecological areas from pollution and disruption. In addition, the departments can build an early warning system for the negative impacts of future LUCC on ecological security patterns, which helps decision-makers prevent the risks in advance.

### 4.3. Limitations and Future Scope

When researchers perform multi-scenario simulations of LUCC, the framework can be combined with time-series forecasting techniques, such as MOP and SD. This facilitates researchers to synthesize the effects of regional policies, current socioeconomic development law, and natural resources on the temporal changes of LUCC. There are still several limitations inherent in our study. On the one hand, spatial heterogeneity exists in the importance of both source and corridor in the regional ESP, which may require relevant managers to design a more considerable protection extent for more important sources and corridors. For example, corridors connecting patches far from each other are more susceptible to external environmental influences, and thus, they need to be protected by wider buffer zones. However, this study's edge width and distance threshold parameters were set mainly with existing studies without thoroughly analyzing the relationship between vegetation, biological activities, and ecological networks in the study area. This may lead to some uncertainty. In future work, more field surveys will be needed to set the corresponding parameters according to the features of the study area.

On the other hand, although we adjusted the transition probability between land types using the MC method for land demand, the land policies related to food security, socioeconomic functions of forests, and urban planning in the CZTMA were not included

in the scenario setting. Therefore, the primary task ahead is to incorporate the SD model that can capture socioeconomic and policy factors into the ESP-PLUS framework. Our study provided valuable thoughts on urban growth optimization and ecological resource protection.

## 5. Conclusions

This study proposed an ESP-PLUS prediction framework. The MSPA and circuit theory were employed to identify the multi-level ESPs and were coupled with the PLUS model to simulate the LUCC under multiple scenarios. The framework widens the approach to ecological security-oriented land change modeling. It provides a reference for curbing the occupation of critical ecological patches and corridors by the buildings and protecting the security of ecosystems.

From 2000 to 2050, the forest accounts for 56%–61% of the study area extent while the buildings account for 3%–14%. The patch number of ecological sources is sixty-one, with a total area of about 5942.5 km$^2$, and forty-six ecological corridors are determined in the CZTMA. In the future, the buildings will keep a net increase. In contrast, the forest and cropland show a net decrease trend. The rate of overall LUCC in different scenarios is UG > HT > CCE > EC. The amounts of building gain and forest loss are enormous in all scenarios, and the largest contributor to the expansion of buildings is forest, followed by cropland. By analyzing landscape patterns, the EC and CCE scenarios could mitigate increased patch fragmentation and shape complexity compared with other scenarios.

The metropolitan area's local governments should optimize the allocation of buildings through integrated planning and government guidance based on the "stock" of existing buildings, promote the integration of ecological protection and governance in the metropolitan area, and strengthen the key supervision of the construction of ecological green centers to reduce the risk of destruction of regional ecological corridors.

**Author Contributions:** Conceptualization, B.Q.; methodology, Z.D.; software, Z.D.; validation, B.Q., Z.D. and H.Z.; formal analysis, Z.D.; investigation, Z.D.; resources, B.Q.; data curation, B.Q. and Z.Z.; writing—original draft preparation, Z.D.; writing—review and editing, Z.D., B.Q., H.Z. and H.X.; visualization, Z.D.; supervision, B.Q. and H.Z.; project administration, B.Q.; funding acquisition, B.Q. and H.Z. All authors have read and agreed to the published version of the manuscript.

**Funding:** This research was funded by the Open Foundation of Hengyang Base of International Centre on Space Technologies for Natural and Cultural Heritage under the auspices of UNESCO (grant 2021HSKFJJ029 and 2022HSKFJJ009), the Natural Science Foundation of Hunan Province (grant 2023JJ30096), Scientific Research Fund of Hunan Provincial Education Department (grant 22B0722), and the Science Foundation of Hengyang Normal University (grant 2021QD02).

**Institutional Review Board Statement:** Not applicable.

**Informed Consent Statement:** Not applicable.

**Data Availability Statement:** Not applicable.

**Conflicts of Interest:** The authors declare no conflict of interest.

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
