# Peer review of "Scenario Simulation of Land Use and Cover under Safeguarding Ecological Security: A Case Study of Chang-Zhu-Tan Metropolitan Area, China"

_forests, doi:10.3390/f14112131_

Round 1
Reviewer 1 Report
The work presented to me for evaluation is interesting, quite well documented, and the literature review was thoroughly done. However, I would like to ask what is new in the presented topic in relation to previous works?
The conclusion should also be improved.
The legend in Figure 63, part e, requires changes.
Author Response
We are very grateful for your valuable comments and suggestions. Based on your suggestions, we have made revisions to the paper. The revised parts of the manuscript are highlighted in red. Our detailed point-by-point response to your comments is provided below.
Comment 1: The work presented to me for evaluation is interesting, quite well documented, and the literature review was thoroughly done. However, I would like to ask what is new in the presented topic in relation to previous works? The conclusion should also be improved.
Response 1: We are very grateful to your comments and constructive suggestions for the manuscript. Human activities are changing the Earth’s system in a way that threatens well-being and development and significantly impacts the environment at the local, regional, and global levels. With the deepening of global change research, land use and cover change (LUCC) have become the core component of global environmental change research. LUCC is both the reason and result of biophysical processes and social economy, greatly influencing climate change, biodiversity, grain yield, and air contamination.
Along with the promotion of China’s ecological civilization construction and the implementation of significant projects for the protection and restoration of essential ecosystems, some scholars have considered nature reserves, biological habitats, ecological red lines, and areas with good ecosystem services as ecological environment constraints in the modeling process from the perspective of ecological protection. In order to provide a scientific basis for mitigating and avoiding the potential impact of future land change on the ecological environment. However, these initiatives still warrant further exploration.
On the one hand, in the past, ecological patches in ecological limiting factors were isolated and disconnected and did not constitute an ecological network system. On the other hand, most of the existing research’s ecological constraints are relatively simple and cannot match the land demand under multiple scenarios.
Currently, constructing a regional ecological security pattern is a primary measure to strengthen favorable ecological processes, control urban sprawl, maintain regional ecological security, and safeguard public well-being. As a banded area in the ecological security pattern, ecological corridors are the core channels connecting the flow of matter, energy information, and other ecological processes. They are vital for species migration and diffusion. Therefore, in the ecological context, it is urgent to change the ecological constraint model in the existing land change modeling. From the perspective of ecological security patterns and the interconnection of ecological patches, ecological areas such as ecological source areas and corridors are used as ecological limiting factors to explore and optimize the future land use pattern. Studies on optimizing land use patterns focus on urban agglomeration, city, county, and other administrative areas. There are few studies on the coordination and balance of ecological security and urban development based on the future land use evolution and ecological security pattern. How can we diagnose the environmental impact of future urban growth on ecological areas by coupling the factors affecting the interconnectivity of ecological networks? Constructing a technical path for optimizing the spatial pattern of metropolitan areas from the perspective of ecological security is an important direction that needs to be explored.
Recently, China has established seven national metropolitan areas since 2020, with breaking the original administrative boundary constraints, aiming:
(i) to optimize the development pattern of the metropolitan regions.
(ii) to promote infrastructure interconnection.
(iii) to create a scientific and innovative industrial system.
(vi) to promote common protection and governance of the ecological environment.
(v) to promote common construction and sharing of public services.
(vi) to promote the construction of a high-standard market system.
In the context of national economic and social development transformation, the policy status of metropolitan area in the implementation of national strategies has been continuously upgraded, and gradually become an important carrier of regional high-quality development. 2022 In April, the National Development and Reform Commission issued the official approval of the Chang-Zhu-Tan Metropolitan Area Development Plan (hereinafter referred to as “the plan”), marking the birth of the first national-level metropolitan area in the central part of China. In April 2022, the National Development and Reform Commission (NDRC) officially approved the Chang-Zhu-Tan Metropolitan Area Development Plan (hereinafter referred to as the “Plan”), marking the birth of the first national metropolitan area in central China. In recent decades, the rural population of Chang-Zhu-Tan metropolitan area has flocked to the cities, and the demand for construction land has continued to rise, resulting in the encroachment of high-quality agricultural land and ecological resources, which has led to a series of problems, such as land degradation, arable land erosion, and water resource contamination, and has posed a challenge to the sustainable development of the metropolitan area's future land space. Influenced by multiple factors at home and abroad, the epidemic has impacted the regional production and industrial chain, and the new downward pressure on the economy has increased. In addition, the development within the metropolitan area is unbalanced and insufficient, and the level of integration and co-location in terms of industrial structure, public facilities, environmental protection, and market system needs to be urgently improved. In line with the requirements of ecological civilization and new urbanization, high-quality socio-economic development remains an intrinsic need for the Chang-Zhu-Tan Metropolitan Area. The Ecological Green Center in the Plan, as a large-scale urban green center in China, is located in the center of the circle and plays an important role in maintaining the ecological function of the metropolitan area and improving the quality of life of the residents but faces the risk of encroachment by the construction of urban and rural facilities in the circle. Therefore, there is an urgent need to explore the future trend of land use change in the circle and its optimization path from the perspective of ecological security.
“However, the studies on multi-scenario simulation of LUCC incorporating ESP in metropolitan areas are still lacking. Hence, the research questions of this study were formulated: (1) How to frame a predictive modeling for LUCC that considers ecological security? (2) What are the LUCC dynamics and their enlightenment under the framework?” (Line 127-132)
Also, we carefully revised the conclusion. Finally, we thank you for the advice and effort to improve the manuscript’s quality.
Comment 2: The legend in Figure 6, part e, requires changes.
Response 2: We appreciate your careful review and suggestion. We have modified the legend issues in the parts of Figure 6e according to your comments. Modified Figure 6 is below:
[Figure 6]
Figure 6. Scenario comparison in the land change process during 2020–2050. Note: different scenarios’ overall changes at interval level (a); scenarios differences at category level (b); ecological area and buildings gains from 2020–2050 under the BAUHT scenario (c); other land categories transition to buildings (d); ecological area loss distribution due to building-buildings growth under the BAUHT scenario (e); HT: historical tendency; UG: urban growth; EC: ecological conservation; CCE: coordinating city development and ecological protection.

Reviewer 2 Report
Land use change, both the analysis of past trends and the anticipation of future developments, is a very important task in creating sustainable development. Such studies are important and necessary.
Multiple methods were used for the study and various development scenarios were developed. This makes the article difficult to read. Depending on the method adopted, more or less optimistic development scenarios were obtained. The conclusions are all too obvious (they confirm general trends). There is a lack of recommendations for the future or an indication of how the scenario can be implemented.
The article highlights ecological content, including ecological corridors. It is worth highlighting the protection of forest land and, in particular, agricultural land with high production values. An overlooked aspect is food security and the social and environmental function of forests in human life.
Author Response
We are very grateful for your valuable comments and suggestions. Based on your suggestions, we have made revisions to the paper. The revised parts of the manuscript are highlighted in red. Our detailed point-by-point response to your comments is provided below.
Comment 1: Multiple methods were used for the study and various development scenarios were developed. This makes the article difficult to read. Depending on the method adopted, more or less optimistic development scenarios were obtained. The conclusions are all too obvious (they confirm general trends). There is a lack of recommendations for the future or an indication of how the scenario can be implemented.
Response 1: We thank the reviewers for the valuable suggestions. We agree with your point of view. We used different models and methods to construct the research framework in the paper. This study aims to develop a coupled ecological security pattern with the PLUS model, aiming to explore the interaction between urban growth, land use and cover change, and the overall security of ecosystems from the perspective of ecological security. At the same time, as the reviewer mentioned, “we may have deficiencies in specific and implementable recommendations,” with which we agree. Therefore, we set up relevant recommendations in Discussion 4.2 and made additions and modifications accordingly.
“Firstly, the local government should firmly grasp the “overall situation”, adhere to the basic guiding principles of coordinated regional development and green ecological priority, and optimize the mechanism of coordinated diagnosis, treatment and restoration of the ecological environment. Secondly, city's planning department should take into account the protection and development of the ecological green heart within the metropolitan area, on the one hand, strengthen the ecological restoration of the fragile areas of the green heart and strictly control the ecological red line area. On the other hand, it is necessary to optimize and revitalize the “stock” of buildings and improve the level of intensive and economical use. Finally, it is necessary to pay attention to the importance of waters and wetlands in maintaining the diversity of ecosystems in the region as well as regulating the hydrological process, and to set up a blue line for the protection of key waters, and to strengthen the management and protection of the shorelines of Xiangjiang River and other lakes, so as to ensure the safety of water re-sources.” (Line 642-654)
In the future, we will conduct the next study based on the official documents of the Chang-Zhu-Tan Metropolitan Area Development Plan. This future study will focus on food security (production land), ecological protection (ecological space), and urban growth (living space). It will incorporate future policies and planning documents into the three spaces to comprehensively predict how the three spaces will evolve in the future and to test whether the policies and plans are optimal or not, which will ultimately serve as a basis for the decision-makers and the natural resource departments to formulate relevant plans and policies.
Comment 2: The article highlights ecological content, including ecological corridors. It is worth highlighting the protection of forest land and, in particular, agricultural land with high production values. An overlooked aspect is food security and the social and environmental function of forests in human life.
Response 2: We thank the reviewer for the helpful comments and suggestions. We agreed the valuable suggestions. After your reminder, we also deeply realize that this manuscript could not focus on the field of food security and social and environmental functions of forests. In the future, we will consider national, regional and local agriculture and food related policies as well as socio-economic functions of forests (e.g., green space, open space, forest products). In section 4.3 of the manuscript, we discussed the limitations of this study and future work. We have also made some changes based on your suggestions (Line 686-689).

Reviewer 3 Report
The paper entitled “Scenario Modeling of Land Use and Cover under Safeguarding Ecological Security: A Case Study of Chang-Zhu-Tan Metropolitan Area, China” reflects the development of applied research in the field of remote sensing and geographical information systems. However, the methods need to be improved. Thus, major changes are recommended.
Comments
1) Keywords – One of the keywords in the title. Please consider replacing it.
2) Line 64, 434 - SLEUTH, CLUE, FLUS, PDU have not been defined. Provide the name prior to the acronym.
3) Line 148 – Please consider using dollars instead of yuan.
4) Figure 1a – Scale is missing.
5) Line 165 – Which are the 6 classes the authors are referring to? Please include its names.
6) Lines 166-167 – fir-level classification should be described with further details.
7) Line 169-170 – Buildings or urban areas?
8) Section 2.2 – The themes the authors used had different special resolutions. Which spatial resolution was used? How did the authors convert the other spatial scales into the one they have used?
9) Line 181, 230, 238, 273 – Reference for software used are missing.
10) Lines 190-192 – How was the null value addressed? Please include further details.
11) Line 228, 229 – Why did the authors use these land use and date as standard?
12) Lines 232, 233 – Definitions and formulas of the beta diversity indices are missing, as well as a literature reference.
13) Lines 253, 254 – What is the justification for the selection of these variables as resistance factors?
14) Table 2 – The variables have no units?
15) Figure 3 – The inclusion in the figure caption of the meaning of the acronyms used would improve its understanding.
16) Line 326 – Table 3 is missing. Is it Table 4?
17) Line 406 – shrunk or decreased?
18) Line 474-485 –It is not clear in the methods were BAU scenario is described. Please clarify.
Moderate editing of English language required
Author Response
We are very grateful for your valuable comments and suggestions. Based on your suggestions, we have made revisions to the paper. The revised parts of the manuscript are highlighted in red. Our detailed point-by-point response to your comments is provided below.
Comment 1: Keywords – One of the keywords in the title. Please consider replacing it.
Response 1: On behalf of all our authors, I am very grateful for your valuable advice. We carefully consider the keywords and the title of the manuscript. Finally, we changed the title of the paper from “Modeling” to “Simulation” and added some additions to the keywords, such as changing “Ecological security” to “Ecological security pattern”. If such corrections do not align with your viewpoint, please do not hesitate to contact us by Forest Editorial Office at any time so that the quality of the manuscript can be better improved before publication.
Comment 2: Line 64, 434 - SLEUTH, CLUE, FLUS, PDU have not been defined. Provide the name prior to the acronym.
Response 2: We agree with your suggestion and have modified the manuscript accordingly. After carefully reviewing a large number of references related to these models, we made the following modifications:
(1) The SLEUTH Model was proposed by Professor Clark of the University of California in the United States, and its abbreviation has also been widely adopted. However, in this first published literature, it is not difficult to find that the abbreviation rule for SLEUTH Model is composed of the first letters of six input factors for land use simulation. These six factors are slope, land use, exclusion layer, urban extent, transportation, hill shade. Thus, the SLEUTH (slope, land-use, exclusion, urban extent, transportation and hill-shade) model was write as “SLEUTH” in references (Silva, E. et al, 2002; Liu, D. et al, 2020; Saxena, A. & Jat, M.K., 2019).
(2) Line 64-We added “the Conversion of Land Use and its Effects” before “CLUE”, According to this work (Verburg, P. H. et al, 2002).
(3) Line 65- We also added “the Future Land Use Simulation” before “FLUS”, According to the paper (Liu, X. et al, 2017).
References:
Silva, E. A.; Clarke, K. C., Calibration of the SLEUTH urban growth model for Lisbon and Porto, Portugal. Computers, Environment and Urban Systems 2002, 26 (6), 525–552.
Liu, D., Clarke, K. C., & Chen, N. Integrating spatial nonstationarity into SLEUTH for urban growth modeling: A case study in the Wuhan metropolitan area. Computers, Environment and Urban Systems 2020, 84, 101545.
Saxena, A., & Jat, M. K. Capturing heterogeneous urban growth using SLEUTH model. Remote Sensing Applications: Society and Environment 2019, 13, 426-434.
Verburg, P. H.; Soepboer, W.; Veldkamp, A.; Limpiada, R.; Espaldon, V.; Mastura, S. S., Modeling the spatial dynamics of regional land use: the CLUE-S model. Environmental Management 2002, 30 (3), 391–405.
Liu, X.; Liang, X.; Li, X.; Xu, X.; Ou, J.; Chen, Y.; Li, S.; Wang, S.; Pei, F., A future land use simulation model (FLUS) for simulating multiple land use scenarios by coupling human and natural effects. Landscape and Urban Planning 2017, 168 (12), 94–116.
Comment 3: Line 148 – Please consider using dollars instead of yuan.
Response 3: Thank you for this comment, and we appreciate your suggestion. Line 149-According to the latest exchange rate, 1.88 trillion yuan has been adjusted to 2.57 hundred billion dollars.
Comment 4: Figure 1a – Scale is missing.
Response 4: We appreciate it very much for this good suggestion, and we have done it according to your ideas. Figure 1a have modified by adding the scale. Also, we have optimized Figure 1 based on the previous figure to help more understand the geographical location of the study area.
Figure 1. Location (a) and administrative divisions (b) of the CZTMA in China.
Comment 5: Line 165 – Which are the 6 classes the authors are referring to? Please include its names.
Response 5: We thank the reviewer for valuable comments and have added the information according technical manual in the data website (http://www.resdc.cn/). Additional information and original description modified to:
“We employed the data product’s the first-level classification of this data product in the study that comprises cropland, forest, grassland, waters, buildings, and unused land (Table 1). The unused land with a small area was classified into grassland, according to the observation of Google Earth images. Finally, we divided land use categories into cropland, forest, grassland, waters, and buildings in the study area.” (Line 171-176)
Table 1. The first-level classification of land use and cover data in the study.
Category |
Description |
Cropland |
Cropland refers to land on which crops are grown, including mature cultivated land, newly opened land, fallow land, swidden agriculture land, and grass-field rotation land; and agricultural fruits and mulberries, which are mainly cultivated with agricultural crops. |
Forest |
Forest refers to forestry land where trees, shrubs, bamboo and coastal mangrove land grow. |
Grassland |
Grassland refers to all types of grassland with a predominantly herbaceous growth and a cover of 5% or more, including scrub grassland with a predominantly pastoral growth and sparse grassland with a canopy density of less than 10%. |
Waters |
Waters refers to natural terrestrial waters and land for water facilities, such as lakes, ponds, reservoirs, and shallows. |
Buildings |
Buildings refers to urban and rural settlements and other industrial, mining, transportation and other land. |
Unused land |
Unused land refers to land that is currently unutilized, including land that is difficult to use, such as sandy land, gobi, saline land, and bare land. |
Comment 6: Lines 166-167 – first-level classification should be described with further details.
Response 6: Thank you for the valuable reminder. In order to make this study better known to the wider audience of the Forests journal, we have described further details of the first-level classification according to the user manual available in the data website (Table 1).
Comment 7: Line 169-170 – Buildings or urban areas?
Response 7: We agree with your suggestion and have changed the word “building” to “buildings” throughout the manuscript.
Comment 8: Section 2.2 – The themes the authors used had different special resolutions. Which spatial resolution was used? How did the authors convert the other spatial scales into the one they have used?
Response 8: Thank you for reminding us to add the process on how to unify the different data resolutions. In the Land Use Simulation Study section, we entered 13 land change drivers and unified the coordinate system and raster spatial resolution of the data for these factors. We utilized the Data Management Tools>>Raster>>Raster Processing>>Resampling tool in ArcGIS Pro 2.5 to standardize all 13 data to a spatial resolution of 30 m that is consistent with the land use and cover data, where the spatial distance data we are using the Euclidean Distance tool in ArcGIS Pro 2.5. This tool itself can set the size of the raster spatial resolution when outputting the results. Therefore, no resampling step is required for this type of data. (See Line 198-204)
Moreover, in the GIS tool, four methods are provided for resampling raster data, namely Nearest, Majority, Bilinear and Cubic. The Nearest Neighbor Allocation method is usually used by default.
Line 195-We added process information: Additionally, the resampling tool in the ArcGIS Pro 2.5 software was used to unify the spatial resolution of the driving factors to 30 m.
Comment 9: Line 181, 230, 238, 273 – Reference for software used are missing.
Response 9: Thank you for the advice. We have corrected the missing references you mentioned in the whole article. We have made references to relevant GIS software.
(1) “ArcGIS Pro software (Version 2.5, Esri, Redlands, CA, USA)” - The reason for this change is that an article published in the issue 5 of the Forests Journal 2023 describes the software in this way (https://doi.org/10.3390/f14050939). We think that perhaps this kind of citation is in line with the requirements of the journal. If there is a more suitable method, we will also make further changes.
(2) “GuidosToolbox software (https://forest.jrc.ec.europa.eu/en/activities/lpa/) (Vogt P. and Riitters K., 2017; Soille P. and Vogt P., 2009)”. This is free download website about Guidos software. The software detail can be shown at the webpage.
(3) We added relevant and useful information - “The Conefor tool and its manual are available for free on the website (http://www.conefor.org/coneforsensinode.html) (Saura, S. & Torné, J., 2009).”
(4) The modification of this point is similar to point (1) – “First, the neighboring source areas were identified using cost allocation and Euclidean allocation functions in ArcGIS software (Version 10.2, Esri, Redlands, CA, USA)”.
References:
Saura, S.; Torné, J., Conefor Sensinode 2.2: A software package for quantifying the importance of habitat patches for landscape connectivity. Environmental Modelling & Software 2009, 24 (1), 135-139. 10.1016/j.envsoft.2008.05.005.
Vogt, P.; Riitters, K., GuidosToolbox: universal digital image object analysis. European Journal of Remote Sensing 2017, 50 (1), 352-361.
Soille, P.; Vogt, P., Morphological segmentation of binary patterns. Pattern recognition letters 2009, 30 (4), 456-459.
Comment 10: Lines 190-192 – How was the null value addressed? Please include further details.
Response 10: The raster data with null values mainly appeared in the DEM data, which could be caused by merging multiple DEM data. Thus, this study adopts the following approach for a dozen null points: first, we used the Con conditional function in the raster calculator in the GIS tool to assign the null value to 1, and other values remain unchanged. In the raster calculator, the specific expression is Con [Isnull (raster), 0, raster].
Land use simulation will not be possible without resolving the null value. This is because any land change prediction model requires consistent spatial projection and resolution of the data as well as no null values in the raster data.
Line 200-204, Added “The null value problem in the DEM data was addressed before input to the PLUS model. For We used the conditional function Con in the raster calculator of the GIS tool to assign the null value to 1, and other values remain unchanged.”
Comment 11: Line 228, 229 – Why did the authors use these land use and date as standard?
Response 11: Morphological Spatial Pattern Analysis (MSPA) refers a sequence of mathematical morphological operators targeted at the description of the geometry and connectivity of the image components (Soille P. and Vogt P., 2009). Based on geometric concepts only, this methodology can be applied at any scale and to any type of digital images in any application field (Soille P. and Vogt P., 2009). Therefore, MSPA method was widely adapted in the identifying ecological network or ecological security pattern field, and so on.
GuidosToolbox is a desktop application with spatial pattern analysis software targeted at image object analysis, generic image processing utilities, exporting raster image (Vogt P. and Riitters K., 2017). The foreground area of a binary image is divided into seven visually distinguished MSPA classes: Core, Islet, Perforation, Edge, Loop, Bridge, and Branch (Table 2).
“Forest and waters have good ecological function and less disturbance of human activities, which is an ideal space for species to survive (Lin, J. et al, 2023; Osewe, E.O. et al, 2022). Buildings and cropland are greatly affected by human activities (Wang, Y. et al, 2022; An, Y. et al, 2021). Moreover, the grassland’s ecological quality in the study area is poor compared with forest and the grassland lacks a better environment for species to forage and rest. Hence, we selected forest and waters as foreground pixels, and other land use and cover categories as background pixels based on the previous works (Zeng, W., et al, 2023).”- Finally, we add the reasons for adopting such a standard. Thanks again for the reviewer's valuable comments. We all believed that this opinion has played a positive role in improving the quality of the Method Section of the manuscript.
Table 2. Foreground classes for morphological spatial pattern analysis.
Foreground classes |
Description |
Core |
interior area excluding perimeter |
Islet |
disjoint and too small to have core |
Loop |
connected to the same core area |
Bridge |
connected to different core areas |
Perforation |
internal area perimeter |
Edge |
external area perimeter |
Branch |
connected at one end to Edge, Perforation. Bridge or Loop |
References:
Soille P. and Vogt P. (2009). Morphological segmentation of binary patterns. Pattern Recognition Letters 30, 4:456-459.
Vogt P. and Riitters K. (2017). GuidosToolbox: universal digital image object analysis. European Journal of Remote Sensing, 50, 1, pp. 352-361.
Zeng, W.; Tang, H.; Liang, X.; Hu, Z.; Yang, Z.; Guan, Q., Using ecological security pattern to identify priority protected areas: A case study in the Wuhan Metropolitan Area, China. Ecological Indicators 2023, 148.
Wang, Y.; Qu, Z.; Zhong, Q.; Zhang, Q.; Zhang, L.; Zhang, R.; Yi, Y.; Zhang, G.; Li, X.; Liu, J., Delimitation of ecological corridors in a highly urbanizing region based on circuit theory and MSPA. Ecological Indicators 2022, 142, 109258. https://doi.org/10.1016/j.ecolind.2022.109258.
An, Y.; Liu, S.; Sun, Y.; Shi, F.; Beazley, R., Construction and optimization of an ecological network based on morphological spatial pattern analysis and circuit theory. Landscape Ecology 2021, 36 (7), 2059-2076.
Lin, J.; Zeng, Y.; He, Y. Spatial Optimization with Morphological Spatial Pattern Analysis for Green Space Conservation Planning. Forests 2023, 14, 1031.
Osewe, E.O.; Niţă, M.D.; Abrudan, I.V. Assessing the Fragmentation, Canopy Loss and Spatial Distribution of Forest Cover in Kakamega National Forest Reserve, Western Kenya. Forests 2022, 13, 2127.
Comment 12: Lines 232, 233 –Definitions and formulas of the beta diversity indices are missing, as well as a literature reference.
Response 12: Beta diversity index is an important method and index in ecology. Thank you very much for the good suggestion. The beta diversity index was not used in the manuscript. This study mainly focuses on the multi-scenario simulation of land use and cover from the perspective of ecological security. For ecological security, we used MSPA, connectivity evaluation and minimum cumulative resistance model to complete the construction of ecological security pattern. According to the previous experimental paradigm of ecological security pattern (Peng J. et al, 2017; Peng J. et al, 2018; Zeng, W. et al, 2023), the first step is to determine the ecological source; The second step is to construct the resistance surface; The third step is to estimate the potential ecological corridors between source areas. Finally, we superimposed the source and corridor, rasterized them, and then coupled them into the simulation model. This is the basic postgraduate idea of this research, and also the novelty of this research. Therefore, we did not adopt the beta diversity index in the modeling process. Of course, this indicator can further enrich the research results and improve the research framework. We hope to give important consideration to the suggestions you mentioned in the application and work of future research projects.
References:
Peng, J.; Zhao, H.; Liu, Y., Urban ecological corridors construction: A review. Acta Ecologica Sinica 2017, 37 (1), 23-30.
Peng, J.; Pan, Y.; Liu, Y.; Zhao, H.; Wang, Y., Linking ecological degradation risk to identify ecological security patterns in a rapidly urbanizing landscape. Habitat International 2018, 71 (1), 110–124.
Zeng, W.; Tang, H.; Liang, X.; Hu, Z.; Yang, Z.; Guan, Q., Using ecological security pattern to identify priority protected areas: A case study in the Wuhan Metropolitan Area, China. Ecological Indicators 2023, 148.
Comment 13: Lines 253, 254 – What is the justification for the selection of these variables as resistance factors?
Response 13: The ecological resistance surface reflects the influence of landscape heterogeneity on ecological processes, and this value is mainly manifested as hindering the spatial movement of species and the flow of material information in the ecosystem. Both natural factors and human activities have a certain influence on ecological processes.
Based on previous studies (Wei, Q. et al, 2022; Li, L. et al, 2022; In, L., et al, 2022; Yi, D., et al, 2022), we consider slope, elevation and distance to buildings as factors for the construction of resistance surfaces. The three factors are continuous variables, and the higher their value, the greater the resistance to ecological processes such as species migration. Under the background of global climate change and urbanization in China, land use and cover type are one of the manifestations of the interaction between social and economic activities and natural environment. It reflects the macro differences in the landscape of the Earth’s surface. Different land use and cover categories have different effects on ecological processes. The higher the ecological function of the type, the less resistance to species migration. In addition, MSPA landscape types are landscape structures generated in the foreground based on forest and waters, and different landscape types also have different resistance to species migration. As the most important part of the seven landscape classes, the core area is assigned a resistance value of 1. In contrast, the background is set to maximum, and other types are assigned according to ecological function importance.
References:
Wei, Q.; Halike, A.; Yao, K.; Chen, L.; Balati, M., Construction and optimization of ecological security pattern in Ebinur Lake Basin based on MSPA-MCR models. Ecological Indicators 2022, 138, 108857.
Li, L.; Huang, X.; Wu, D.; Wang, Z.; Yang, H., Optimization of ecological security patterns considering both natural and social disturbances in China's largest urban agglomeration. Ecological Engineering 2022, 180, 106647.
In, L.; Xu, Q.; Yi, J.; Zhong, X., Integrating CVOR-GWLR-Circuit model into construction of ecological security pattern in Yunnan Province, China. Environ Sci Pollut Res Int 2022, 29 (54), 81520-81545.
Yi, D.; Guo, X.; Han, Y.; Guo, J.; Ou, M.; Zhao, X., Coupling Ecological Security Pattern Establishment and Construction Land Expansion Simulation for Urban Growth Boundary Delineation: Framework and Application. Land 2022, 11 (3), 359.
Comment 14: Table 2 – The variables have no units?
Response 14: Thank you for this suggestion. We have added unit information based on different elements. The table is revised below:
Table 3. Design and weight of resistance surface factors.
Resistance factor |
Weight |
Resistance value |
||||
1 |
3 |
5 |
7 |
9 |
||
Slope |
0.17 |
<3° |
3°–8° |
8°–15° |
15°–25° |
>25° |
Elevation |
0.07 |
<50 m |
50–150 m |
150–250 m |
250–350 m |
>350 m |
Land use category |
0.37 |
Forest |
Grassland |
Cropland |
Waters |
Buildings |
MSPA landscape type |
0.17 |
Core |
Loop, Bridge |
Islet, Branch |
Perforation, Edge |
Background |
Distance to buildings |
0.22 |
>1500 m |
1000–1500 m |
500–1000 m |
200–500 m |
<200 m |
Comment 15: Figure 3 – The inclusion in the figure caption of the meaning of the acronyms used would improve its understanding.
Response 15: We have added the relevant information in the captions of the figures where the abbreviations appear according to your comments.
Comment 16: Line 326 – Table 3 is missing. Is it Table 4?
Response 16: Thanks for the heads up. line 326 should appear as table 4. this was our oversight. We have correctly updated it according to the latest table dynamics of the manuscript. Thank you again for your valuable suggestion that helped us avoid such an error.
Comment 17: Line 406 – shrunk or decreased?
Response 17: We carefully examined this sentence in Line 406 and found it to be ambiguous. Because it does not make it clear whether there is an increase or a decrease. In fact, what we want to convey is that under the HT scenario, the buildings’ area in 2050 increases by 71% compared to 2020. Therefore, we have modified the sentence - “Under the HT scenario, the building's area will be 2439.4 km2 by 2050, an increase of 71% compared to 2020.”
Comment 18: Line 474-485 –It is not clear in the methods were BAU scenario is described. Please clarify.
Response 18: Thank you very much for suggesting. Indeed, the occurrence of this error is the result of our carelessness. Hence, it has caused you trouble in reviewing the manuscript. On behalf of all our authors, I would like to apologize to you. The “BAU scenario” is the manuscript’s original description. Before submitting the manuscripts, we revised all the scenario designations, but maybe some of the original words were not corrected in time. The BAU scenario refers to the historical tendency scenario in the current manuscript. We have changed BAU to HT, PDU to UG, PPE to EC and BUE to CCE by your comments. Thank you again for your valuable contribution to the quality improvement of the manuscript.
Finally, we worked with colleagues to carefully revise the entire manuscript according to the English check.

Reviewer 4 Report
It is a very interesting and well prepared article concerning crucial problem as regards biodiversity development and protection of natural ecosystems: land use and cover planning and modeling as the important factor depending sustainable development and ecologization of the area (rural and urban).
The Authors have undertaken the scientific trial to construct the methodical approach enabling monitoring and optimization patterns of land use, with elements of evaluation and prediction of these changes in future.
The Authors were successful, though used well-known methods, but construction of the methodical approach is partly innovative. Figure 2 shows the framework of the study and is very helpful for other researchers to repeat the study or implement this methodical approach in an another study.
Results are interesting, but relatively easy to predict. However, readers will be very curious to follow the paper, because the problem and topic are extremely important now - in the period of the great threat for our Planet, including consequences of anthropogenic climate change.
In general, presented study expands the knowledge in the field of the issued problem, including the possible methodical aproach to use.
Detailed notes:
Ad ABSTRACT - please, formulate better the aim of the study
Line 125 - rather "formulated", not "formed".
Summerizing, the paper is good and worth publishing after very minor revisions.
the same as above
Author Response
We are very grateful for your valuable comments and suggestions. Based on your suggestions, we have made revisions to the paper. The revised parts of the manuscript are highlighted in red. Our detailed point-by-point response to your comments is provided below.
Comment 1: Ad ABSTRACT - please, formulate better the aim of the study.
Response 1: We co-authors agree with the suggestion to further modify the purpose of the study in the abstract. We have thoroughly and carefully reviewed the abstract and revised the purpose of the study. The revised version is summarized as follows:
Scenario-based simulation in land use and cover change (LUCC) is a practical approach to maintaining ecological security. Many studies generally set constraints of LUCC utilizing eco-logical patches but without consideration of corridors connecting these patches. Here, we con-structed a framework to balance urban growth and ecological security by integrating ecological security patterns (ESP) into the PLUS model. This study selected Chang-Zhu-Tan Metropolitan Area (CZTMA) in central China as a typical case. Specifically, coupling quantitative demand with spatial constraints of multiple levels of ESP, this study designed four scenarios, including his-torical tendency (HT), urban growth (UG), ecological conservation (EC), and coordinating city development and ecological protection (CCE). Then, the transformations and landscape patterns of LUCC were analyzed to evaluate the future land change from 2020 to 2050. The results show sixty-one key ecological sources in the CZTMA, mainly in higher-elevation forested areas. For-ty-six ecological corridors were estimated using circuit theory. The buildings expansion was driven by accessibility to transportation and government locations and will contribute to the loss of forest and cropland in the future. The feature of different scenarios in alleviating the increasing fragmentation of patches and reducing the loss amount of ecological land showed EC>CCE>HT>UG. The study developed the EPS-PLUS framework and its modeling idea, which has the potential to be applied in other regions. This extension would assist decision-makers and urban planners in formulating sustainable land strategies that effectively reconcile eco-environmental conservation with robust economic growth, achieving a mutually beneficial outcome.
Comment 2: Line 125 - rather "formulated", not "formed".
Response 2: Thank you very much for the valuable advice. We all agree that “formulated” is more appropriate than “formed”. It has been modified.
Finally, we worked with colleagues to carefully revise the entire manuscript according to the English check.

Round 2
Reviewer 3 Report
The manuscript has improved in the second version and the questions were addressed by the authors. Thus, publication is recommended.
Minor editing of English language required